# Galectin-3/Gelatin Electrospun Scaffolds Modulate Collagen Synthesis in Skin Healing but Do Not Improve Wound Closure Kinetics

**DOI:** 10.3390/bioengineering11100960

**Published:** 2024-09-25

**Authors:** Karrington A. McLeod, Madeleine Di Gregorio, Dylan Tinney, Justin Carmichael, David Zuanazzi, Walter L. Siqueira, Amin Rizkalla, Douglas W. Hamilton

**Affiliations:** 1Faculty of Engineering, School of Biomedical Engineering, University of Western Ontario, London, ON N6A 3K7, Canadaarizkall@uwo.ca (A.R.); 2Department of Anatomy and Cell Biology, University of Western Ontario, London, ON N6A 5C1, Canada; dtinneyd@uwo.ca (D.T.); jcarmi25@uwo.ca (J.C.); 3Biochemistry Schulich School of Medicine and Dentistry, University of Western Ontario, London, ON N6A 5C1, Canada; dmachad@uwo.ca (D.Z.); walter.siqueira@usask.ca (W.L.S.); 4College of Dentistry, University of Saskatchewan, Saskatoon, SK S7N 5E5, Canada; 5School of Dentistry, Schulich School of Medicine and Dentistry, University of Western Ontario, London, ON N6A 5C1, Canada

**Keywords:** wound chronicity, macrophage, keratinocytes, lectins, matricellular proteins, arginase-I, keratin 17, re-epithelialization

## Abstract

Chronic wounds remain trapped in a pro-inflammatory state, with strategies targeted at inducing re-epithelialization and the proliferative phase of healing desirable. As a member of the lectin family, galectin-3 is implicated in the regulation of macrophage phenotype and epithelial migration. We investigated if local delivery of galectin-3 enhanced skin healing in a full-thickness excisional C57BL/6 mouse model. An electrospun gelatin scaffold loaded with galectin-3 was developed and compared to topical delivery of galectin-3. Electrospun gelatin/galectin-3 scaffolds had an average fiber diameter of 200 nm, with 83% scaffold porosity approximately and an average pore diameter of 1.15 μm. The developed scaffolds supported dermal fibroblast adhesion, matrix deposition, and proliferation in vitro. In vivo treatment of 6 mm full-thickness excisional wounds with gelatin/galectin-3 scaffolds did not influence wound closure, re-epithelialization, or macrophage phenotypes, but increased collagen synthesis. In comparison, topical delivery of galectin-3 [6.7 µg/mL] significantly increased arginase-I cell density at day 7 versus untreated and gelatin/galectin-3 scaffolds (*p* < 0.05). A preliminary assessment of increasing the concentration of topical galectin-3 demonstrated that at day 7, galectin-3 [12.5 µg/mL] significantly increased both epithelial migration and collagen content in a concentration-dependent manner. In conclusion, local delivery of galectin 3 shows potential efficacy in modulating skin healing in a concentration-dependent manner.

## 1. Introduction

Skin plays several important physiological roles including water regulation and thermoregulation as well as acting as a barrier against physical, chemical, and biological stresses [1,2]. Despite a robust healing response post-injury, due to diabetes, pressure, and vascular insufficiency, skin wounds can fail to proceed through the normal reparative process and if unresolved after three months, are classified as a chronic (non-healing) wound [3,4]. Non-healing wounds exhibit multiple etiologies, but are associated primarily with prolonged expression of pro-inflammatory cytokines, such as interleukin-1 (IL-1), interleukin-6 (IL-6), and tumor necrosis factor-α (TNF-α), which potentiate the immune response and cause persistent local inflammation [5,6]. Strategies that modify cellular processes associated with inflammation including monocyte recruitment as well as macrophage phenotype and initiate the proliferative phase of healing could have a significant impact on stalled healing and facilitate closure. 

Matricellular proteins (MPs) are non-structural components of the extracellular matrix that are upregulated in wound healing and pathological processes [7,8]. During skin healing, MPs including thrombospondins, periostin, and CCNs act spatially and temporally to control different elements of inflammation and the proliferative phases of healing [9]. As a class of MPs, galectins have received focus due to their potential role in influencing inflammatory processes [10]. Galectin-3, in particular, has been implicated in the regulation of inflammatory and immunomodulatory processes [11], including neutrophil activation, as well as monocyte and macrophage migration [12], clearance of neutrophils [13], and regulation of macrophage polarization [14]. Galectin-3 can activate neutrophils in a dose-dependent manner, through a process involving its carboxyl-terminal domain (CRD) [15]. In monocytic lineages, galectin-3 induces monocyte migration in vitro, with concentration-dependent effects evident in relation to chemotaxis and chemokinesis [12]. Galectin-3-mediated migration of both monocytes and macrophages is increased in the presence of a fibronectin matrix, suggesting that galectin-3 may mediate the linkage of these cells to fibronectin [16]. Another prominent role of the macrophage in inflammation, particularly in tissue repair, is to rid the wound of neutrophils by ingesting them and inducing their apoptosis [17,18]. Galectin-3 can act as an opsonin, linking the phagocytic macrophages to the apoptotic neutrophils directly [13]. The clearance of neutrophils by macrophages is a pivotal step in the suppression of acute inflammation and induction of the late inflammatory phase (M2 macrophage polarization) and proliferative phase of healing. Interestingly, in macrophages derived from galectin-3 deficient mice, IL-4/IL-13-induced M2 macrophage polarization was inhibited, suggesting that galectin-3 also modulates phenotypes associated with regenerative macrophage activation [14].

With respect to skin healing specifically, analysis of the galectin-3 knockout mouse highlighted a migratory defect in keratinocytes that results in a minor delay in the closure of excisional skin wounds compared to wild-type mice [19,20]. This defect results from deficient epidermal growth factor receptor (EGFR) endocytosis and recycling. Specifically, galectin-3 controls receptor cycling by binding to alpha-1,3/1,6-mannosyltransferase interacting protein X (ALIX) [19]. Galectin-3 has also been implicated in the regulation of fibrotic processes and myofibroblast differentiation [21,22], processes required for the proliferative phase of healing [9]. Analysis of galectin-3 delivery in skin healing has, however, led to conflicting results on its efficacy [23,24].

Based on our previous analysis of the knockout mouse [20] and the described roles of galectin-3 in inflammatory processes and epithelial migration [10,15,16,25], we hypothesized that local delivery of galectin-3 could represent a therapeutic strategy to modulate inflammation and re-epithelialization. Two delivery methods were investigated; electrospinning of galectin-3 with gelatin to produce a fibrous scaffold implanted at the time of wounding and secondly, a topical delivery of galectin-3 applied every day to the wound bed. We conclude that topical application of galectin-3 could represent a possible adjunct for altering macrophage polarization, epithelial migration, and collagen production in a concentration-dependent manner.

## 2. Materials and Methods

### 2.1. Electrospinning

A polymer solution consisting of Type B Bovine gelatin (Sigma-Aldrich, St. Louis, MO, USA) dissolved in 40% *v*/*v* acetic acid (Thermo Fisher Scientific, Waltham, MA, USA) was passed through a plastic 1 cc syringe (Terumo, Shibuya, Tokyo, Japan) and 20 gauge blunt-tip stainless steel needle using a syringe pump (VWR International, Radnor, PA, USA). The needle was connected to a high-voltage DC power supply (Gamma High Voltage Research, Ormond Beach, FL, USA) and a grounded stainless steel rotating mandrel. The mandrel speed was held constant at 100 revolutions per minute (RPM). Gelatin Sigma-Aldrich, St. Louis, MO, USA) was dissolved in acetic acid and the concentration, flowrate, and collector distance were varied as outlined in Figure 1A. Acetic acid was selected to dissolve gelatin based on prior analysis and to reduce the potential cytotoxicity of other solvents [26]. To fabricate each gelatin/galectin-3 scaffold, 5, 10, or 20 μL of a 1 mg/mL solution of recombinant human galectin-3 (R&D Systems, Minneapolis, MN, USA) in phosphate-buffered saline (PBS) was added to 0.75 mL of the polymer solution (gelatin and acetic acid) for a final concentration of 6.7, 12.5, and 25 μg/mL, respectively. In gelatin scaffolds, 5 μL of PBS was added to the electrospinning solution. Scaffolds were produced by electrospinning for 1.5 h using a total volume of 0.75 mL of each solution. Scaffolds were then crosslinked in a glass desiccator (VWR International) containing drierite (W.A. Hammond Drierite Co. Ltd., Xenia, OH, USA) using the vapor from a 5 mL solution consisting of 1.5% *v*/*v* glutaraldehyde (GTA) (Sigma-Aldrich, St. Louis, MO, USA) in anhydrous ethyl alcohol (Commercial Alcohols, Brampton, ON, USA), similar to the methods of Zha et al. [27]. The desiccator was held under vacuum for 20 min and scaffolds were left in the sealed desiccator for 48 h to ensure sufficient crosslinking had taken place. Following crosslinking, scaffolds were stored in separate sealed plastic containers with desiccant at 2–8 °C.

### 2.2. Assessment of Fiber Morphology

Three separate scaffolds (*N* = 3) were electrospun at each set of conditions listed in Figure 1A. One circular sample (8 mm diameter) of spun fibers was collected per scaffold using a biopsy punch (Integra Miltex, York, PA, USA). Samples were mounted on aluminum stubs and sputter-coated with osmium. Images were taken for each sample using a scanning electron microscope (LEO 1530; Carl Zeiss, Oberkochen, Germany) at 2 kV and one of three magnifications: 1000×, 3000×, or 5000×. Using ImageJ software version 1.51 (National Institutes of Health, Bethesda, MD, USA), the diameter of 250 fibers (*N* = 3 independent scaffold batches) was measured from 5 separate images from each batch at the same magnification.

### 2.3. Mercury Porosimetry

Mercury porosimetry was used to assess the porosity of the refined gelatin scaffolds. For each test, two scaffolds were electrospun from the same polymer solution for 1.5 h using the parameters outlined in Figure 1A. The scaffolds were subsequently crosslinked for 48 h in 1.5% GTA vapor as previously described. Both scaffolds, measuring approximately 4 × 10 cm, were then removed from the aluminum foil, folded, and placed together in a 5cc stem, which was loaded into the AutoPore IV 9500 mercury porosimeter (Micrometrics Instrument Corp., Norcross, GA, USA). The porosimeter generated a pore size distribution, calculated the scaffold porosity, and calculated the average pore diameter (pore size) of the scaffolds. 

### 2.4. Mass Spectrometry

Mass spectrometry was used to validate the presence of galectin-3 within the scaffold. Prior to conducting mass spectrometry, three gelatin/galectin-3 scaffolds (6.7 μg/mL) were blend-electrospun and crosslinked as described in Section 2.1. One piece of each scaffold measuring approximately 9 cm^2^ was cut from each scaffold for mass spectrometry. Each sample was then denatured and reduced with 4 M urea, 10 mM dithiothreitol (DTT), and 50 mM NH4HCO_3_ (pH 7.8), tryptic digested (2% (*w*/*w*) sequencing-grade trypsin (Promega, Madison, WI, USA)), desalted with C-18 ZipTip^®^ Pipette Tips (Millipore, Billerica, MA, USA), and subjected to mass spectrometry analysis (LC-ESI-MS/MS) using a LTQ-Velos (Thermo Scientific, San Jose, CA, USA) according to the methods described by Moffa et al. [28]. The processing of samples and mass spectrometry were conducted by the Siqueira Laboratory.

### 2.5. Scaffold Preparation for Cell Culture and Animal Studies

Prior to cell culture studies, scaffolds were removed from the sealed plastic containers and each scaffold was quenched in 50 mL of 0.1 M glycine (Sigma-Aldrich, St. Louis, MO, USA) for 1 h to remove residual GTA. Following quenching, three 15-min PBS rinses were conducted, and the scaffolds were left in PBS at 4 °C overnight. For sterilization, scaffolds were placed under ultra-violet (UV) light for 60 min. 

### 2.6. Cell Isolation

Isolation and experimental use of cells derived from human tissue were approved by the Western University Review Board for Health Sciences Research Involving Human Subjects and were in accordance with the 1964 Declaration of Helsinki. Human dermal fibroblasts (HDF) were isolated from skin removed under informed consent from patients undergoing elective lower limb amputation at Victoria Hospital, London, Ontario, according to our standard protocols [29,30,31,32,33]. Human dermal fibroblast populations, isolated from 3 different individuals, were used up to passage 6 and were maintained in Dulbecco’s Modified Eagle Medium (DMEM; Thermo Fisher Scientific, Burlington, ON, Canada) supplemented with 10% fetal bovine serum (FBS) (Gibco Life Technologies, Burlington, ON, Canada) and 1% antibiotics and antimycotic (AA) solution (Gibco Life Technologies, Burlington, ON, Canada) in a humidified environment at 37 °C, 5% CO_2_. The media is changed every 2–3 days.

### 2.7. Adhesion Assay

Scaffolds were punched into circular samples using a 6 mm diameter biopsy punch and placed into a 96-well cell culture plate (BD FalconTM, Franklin Lakes, NJ, USA). HDFs were suspended in serum-free Dulbecco’s Modified Eagle Medium (DMEM) (Gibco, Carlsbad, CA, USA) supplemented with 1% antibiotic-antimicotic (AA) (100 U/mL penicillin, 100 μg/mL streptomycin and 0.25 μg/mL amphotericin B; Gibco) and seeded at a concentration of 3 × 10^4^ cells/mL. HDFs seeded on 96 well plates in the absence of scaffolds served as baseline controls. One hour following seeding, the media was removed and wells were rinsed three times with PBS (Gibco) to remove non-adherent cells and residual media. Scaffolds were transferred to a 500 μL microcentrifuge tube (Port City Diagnostics, Wilmington, NC, USA) and frozen at −80 °C. Similarly, the 96-well plates containing the TCP control HDF samples were stored at −80 °C until they were assayed. The cell number on TCP, gelatin, and gelatin/galectin-3 scaffolds were measured using the CyQUANT^®^ Proliferation Assay Kit (Molecular Probes, Carlsbad, CA, USA) according to the manufacturer’s protocols. Briefly, samples were thawed at room temperature, and 200 μL of CyQUANT^®^ GR dye/cell lysis buffer was added to each microcentrifuge tube or well of the 96-well plate for 5 min at room temperature while subjected to vortexing. Samples were covered in aluminum foil during incubation. The supernatant from each sample was collected and transferred to a flat-bottom black 96-well microplate. Serial dilutions of the cell pellets of known number in CyQUANT^®^ GR dye/cell lysis buffer were also transferred to the 96-well microplate to create a standard curve. The fluorescence of each sample was measured using a Safire2 microplate reader (Tecan, Männedorf, Switzerland) at an excitation wavelength of 480 nm and an emission wavelength of 520 nm. The cell number for each experimental condition was generated from the standard curve. 

### 2.8. Proliferation Assay

Scaffolds were punched into circular samples using a 6 mm diameter biopsy punch and placed into a 96-well cell culture plate (Becton Dickinson, Falcon^TM^, Mississauga, ON, Canada). HDFs were seeded into wells at a density of 3.3 × 10^4^ cells/mL and were cultured for 1, 7, 10, or 14 days in DMEM (Thermo Fisher Scientific Inc., Etobicoke, ON, Canada) supplemented with 10% fetal bovine serum (FBS) (Thermo Fisher Scientific Inc.) and 1% AA. Split media changes were performed (75 μL) every 2 days. At each experimental time point, the media was removed, and wells were rinsed three times with PBS to remove non-adherent cells and residual media. Scaffolds were transferred to 500 μL microcentrifuge tubes and both scaffolds and 96-well plates containing the TCP control HDF samples were stored at −80 °C until assayed. A cell pellet containing 2 × 10^5^ cells was also frozen at −80 °C. The cell number at each time point was determined using the CyQUANT^®^ proliferation assay kit according to the manufacturer’s protocols.

### 2.9. Assessment of Scaffold Biocompatibility

Scaffolds were punched into circular samples using a 6 mm diameter biopsy punch and placed into a 96-well cell culture plate (BD Falcon^TM^). HDFs were seeded into wells at a density of 3.3 × 10^4^ cells/mL and were cultured for 3 and 7 days in DMEM supplemented with 10% FBS (Gibco), 1% AA, and 50 μg/mL ascorbic acid. The media was changed every 2 days. At each experimental time point, cells were fixed in 4% paraformaldehyde (PFA) in PBS for 5 min. Three rinses with PBS were conducted followed by treatment with 0.1% Triton X-100 in PBS for 5 min to permeabilize the cell membranes. Cells were rinsed again in PBS three times, followed by blocking with 1% bovine serum albumin (BSA) in PBS at 4 °C overnight. Scaffolds were incubated with primary antibodies against fibronectin (sc-8422; Santa Cruz Biotechnology, Dallas, TX, USA) diluted at 1:100 in 1% BSA in PBS for one hour at room temperature and were rinsed three times with PBS for 5 min. Scaffolds were incubated for 90 min at room temperature with Indodicarbocyanine (Cy5) Goat Anti-Mouse IgG antibody (115-175-166; Jackson Immuno Research Laboratories, West Grove, PA, USA) at a 1:200 dilution. Negative controls were prepared without the addition of the primary antibody. Following incubation, scaffolds were washed three times for 5 min in PBS and mounted on glass coverslips using Vectashield Mounting Medium (Vector Laboratories, Burlington, ON, Canada) containing 4′,6-diamidino-2-phenylindole (DAPI). Coverslips were sealed with clear nail enamel. Samples were analyzed with an Axio Observer Z1 fluorescence microscope (Carl Zeiss, Oberkochen, Germany) using the appropriate filters. Negative controls were imaged to set the threshold values for the detection of fluorescence. 

### 2.10. Animal Surgeries and Wound Closure Kinetics Study

All animal procedures were performed in compliance with protocols approved by the University Council on Animal Care at Western University. Six wild-type (WT) (C57BL/6J; 000664) mice (Jackson Laboratory; Sacramento, CA, USA) were used for each experiment. All mice were age- and sex-matched and were 11 weeks of age at the time of surgery. Prior to surgery, all mice were given 0.05 mg/kg of buprenorphrine as a pre-emptive analgesic. Animals were then anesthetized using isoflurane, fur was removed from the surgical site, and povidone-iodine was used to clean the area. Four full-thickness wounds measuring 6 mm in diameter were then created using a sterile biopsy punch. For experiment 1, each wound was assigned one of four treatment conditions: empty (control wound), gelatin scaffold, a gelatin scaffold made using 3.3 μg/mL galectin-3, or a gelatin scaffold made using 6.7 μg/mL galectin-3 (*N* = 6 wounds for each treatment group). For experiment 2, each wound was assigned one of four treatment conditions: empty (control wound), a gelatin scaffold made using 6.7 μg/mL galectin-3, topical galectin-3 [12.5 µg/mL], and finally, topical galectin-3 [25 µg/mL] (*N* = 6 wounds for each treatment group). Treatments in both experiments were rotated clockwise in each mouse to eliminate positional effects on wound healing. Mice were injected with 0.05 mg/kg of buprenorphrine again following surgery. Tissue samples of the wounds were harvested immediately afterward and were fixed in 10% neutral buffered formalin (Sigma-Aldrich) for 24 h at 4 °C, transferred to 70% ethanol (Commercial Alcohols, Chatham, ON, Canada), and were paraffin-embedded. Serial 5 μm sections were taken from the center of the wounds. To calculate wound closure kinetics, all mice were imaged using a digital camera (Canon, Tokyo, Japan) with photos taken of the wounds every second day post-surgery. A ruler was included in each image so that the measurements of the wound area could be standardized. Image J software version 1.51 (National Institutes of Health, Bethesda, MO, USA) was used to calculate the wounded area at each time point [20]. 

### 2.11. Histological Analysis of Re-Epithelialization, Macrophage Polarization, and Collagen Production

Tissue samples from each experimental condition were harvested at designated time points and were fixed in 10% neutral buffered formalin (Sigma-Aldrich, St. Louis, MO, USA) for 24 h at 4 °C, transferred to 70% ethanol (Commercial Alcohols), and paraffin-embedded. Serial 5 μm sections were taken from the center of the wounds. Image J software (National Institutes of Health) was used to calculate the wounded area at each time point [19,20,31,34]. Masson’s Trichrome staining, conducted by the Pathology Department at the London Health Sciences Centre, was used to visualize tissue structure, cellular infiltration, and re-epithelialization. Sections were imaged with a Leica DM100 light microscope (Leica, Wetzlar, Germany). Analysis was conducted on Masson’s Trichrome stained sections using ImageJ software to measure the length of the epithelial tongue and the thickness of the epithelium [20]. To visualize collagen deposition in the granulation tissue, sections were stained using the Picro Sirius Red Stain Kit (ab150681; Abcam, Waltham, MA, USA), which was visualized under brightfield and polarization optics. 

Immunohistochemical staining for inducible nitric oxide synthase (iNOS) and arginase I was performed to visualize M1 and M2 macrophage populations. Sections were rehydrated, rinsed with PBS for 5 min, and subjected to enzymatic antigen retrieval for 15 min at 37 °C. Samples were rinsed again in PBS for 5 min at room temperature and blocked using 10% horse serum in PBS for 30 min at room temperature in a humidified chamber. Sections were then incubated in primary goat antibodies against arginase I (sc-18354; Santa Cruz Biotechnology, Dallas, TX, USA) diluted at 1:100 in 10% horse serum and rabbit antibodies against iNOS (ab3523; Abcam, Cambridge, UK) diluted at 1:25 in 10% horse serum overnight at 4 °C. To assess keratinocyte migration, sections were labeled with specific antibodies to keratin 17 (ab109725; Abcam) diluted at 1:100 in 10% horse serum overnight at 4 °C. Sections were rinsed in PBS and incubated with secondary antibodies at a dilution of 1:500 in horse serum for one hour at room temperature, while protected from light. The antibodies included an Alexa Fluor 647 anti-rabbit antibody (ab150079; Abcam) and an Alexa Fluor 488 anti-goat antibody (ab150089; Abcam). Hoechst 33342 (Trihydrochloride Trihydrate; Thermo Fisher Scientific, Etobicoke, ON, Canada) was also added at a dilution of 1:1000 to label nuclei. Sections were rinsed in PBS to remove unbound antibodies and were mounted using Immuno-Mount (Thermo Fisher Scientific, Etobicoke, ON, Canada) mounting medium. Coverslips were sealed with clear nail enamel. Sections were imaged using an Axio Observer Z1 fluorescence microscope (Carl Zeiss, Oberkochen, Germany) using the appropriate filters. Negative controls were sectioned and stained without the addition of primary antibodies. These negative control slides were imaged to set the threshold values for the detection of fluorescence. ImageJ software was used to quantify the number of arginase I-positive macrophages in the wound bed in WT mice at day 7 (*N* = 3 independent mice, *n* = 3 sections per wound) (National Institutes of Health).

## 3. Results

### 3.1. Influence of Electrospinning Parameters on Fiber Diameter and Scaffold Morphology

To determine the influence of electrospinning parameters on the fibers formed, three different concentrations of gelatin, three solution flowrates, and three collector distances were investigated (Figure 1A). To determine the influence of the flowrate and needle-to-collector distances, these parameters were varied while the concentration of gelatin was held constant at 20% weight. Statistical analysis revealed that at each flowrate assessed, there were no significant differences in the fiber diameter when the collector distance was increased (Figure 1B; *p* > 0.05). Additionally, at each collector distance, there were no significant differences in the fiber diameter when the flowrate was increased or the collector distance was increased (Figure 1C; *p* > 0.05). To determine the influence of gelatin concentration on the fiber diameter, both the solution flowrate and the collector distance were held constant while the gelatin concentration was increased. As the concentration of gelatin was increased, the resulting fiber diameter increased (Figure 2A–I). Fiber morphology was assessed in scanning electron microscopy (SEM) images taken at each concentration of gelatin. At the 20% weight concentration of gelatin, SEM analysis revealed the presence of beaded fibers in the electrospun fiber mat (Figure 2A,D). In the fibrous mats electrospun using a 25% weight solution of gelatin, SEM showed that the mats contained various web-like and ribbon-like fibers (Figure 2B,E). Scaffolds electrospun using 30% weight gelatin consisted mainly of ribbon-like fibers, although the relative abundance of cylindrical and ribbon-like fibrils was not quantified (Figure 2C,F). Scaffolds electrospun using a concentration of 30% weight gelatin had a significantly larger mean fiber diameter than fibers electrospun using 20% weight and 25% weight gelatin (Figure 2G–I; *p* < 0.05). Differences in the mean fiber diameter between 20% weight and 25% weight gelatin were not statistically significant. Increases in fiber diameter corresponding to increases in gelatin concentration were apparent in the fiber diameter distributions (Figure 2 G–I). When 30% weight gelatin was used, most fibers ranged from 500 to 1500 nm in diameter. At 25% weight, most of the fibers had a diameter within 300–500 nm, with the distribution of fiber diameter being even smaller at 20% weight gelatin, measuring between 100 and 200 nm.

To determine whether increasing the gelatin concentration above 20% weight could eliminate beaded fibers while maintaining fiber diameters within the 100–200 nm range, the gelatin concentration was increased to 21% weight and SEM was conducted to determine the morphological characteristics as well as measure the resulting mean fiber diameter. The resulting mean fiber diameter was 224.6 ± 13.39 nm and SEM revealed that there were no beads within the fiber mat (Figure 2J,K). The frequency distribution obtained from this sample revealed that fiber diameters ranged from roughly 100–300 nm and fibers in the range of 230–250 nm were most frequently measured.

### 3.2. Scaffold Porosity Is Sufficient for Cell Growth

Mercury porosimetry was conducted to evaluate scaffold porosity and to determine whether scaffold pore sizes would be sufficient to allow cell infiltration. Analysis revealed that scaffolds were 83.08 ± 4.06% porous with an average pore diameter of 1.15 ± 0.77 μm (*N* = 3). A representative graph showing the pore size distribution is shown in Figure 2L. The scaffolds contained pores with two predominant size distributions ranging from 0.3–0.8 μm and 30–50 μm in diameter. 

### 3.3. Detection of Galectin-3 in Scaffolds

To ensure that the blend electrospinning method resulted in scaffolds containing recombinant human galectin-3 (Sequence shown in Figure 3A), mass spectrometry was conducted on crosslinked samples of scaffolds (Figure 3B,C). Four peptide sequences from recombinant human galectin-3 were identified, verifying its presence within the scaffolds. Identified sequences were run using the Basic Local Alignment Search Tool (BLAST) in the Universal Protein Resource (UniProt) database, which showed that each detected peptide sequence aligned to a specific sequence contained within the CRD of human recombinant galectin-3 and matched with 100% sequence identity to human galectin-3 protein (UniProt Accession # P17931) (Figure 3C).

### 3.4. Scaffolds Increase the Initial Adhesion of Human Dermal Fibroblasts

Human dermal fibroblasts were seeded onto tissue culture plastic, gelatin scaffolds, and gelatin/galectin-3 scaffolds (6.7 μg/mL). The cells adhered to all surfaces within one hour (Figure 4A). Significantly more cells were detected on gelatin scaffolds relative to the tissue culture plastic (*N* = 3 independent experiments, *n* = 3 replicates per experiment, *p* < 0.01). Similarly, significantly more cells attached to gelatin/galectin-3 scaffolds than tissue culture plastic (*N* = 3 independent experiments, *n* = 3 replicates per experiment, *p* < 0.001). However, no significant differences in cell number were detected between the gelatin and gelatin/galectin-3 scaffolds at one hour following seeding (*N* = 3 independent experiments, *n* = 3 replicates per experiment, *p* > 0.05).

### 3.5. Scaffolds Support the Proliferation of Human Dermal Fibroblasts

To assess increases in human dermal fibroblast numbers, cell numbers were quantified at days 1, 7, 10, and 14 post-seeding. Cell numbers increased over a 14-day period when cultured on tissue culture plastic, the gelatin scaffold, and the gelatin/galectin-3 scaffold (Figure 4B). There were no significant differences in cell numbers between the three conditions at each time point assessed (*N* = 3 independent experiments, *n* = 4 replicates per experiment, *p* > 0.05).

### 3.6. Scaffolds Support the Production of Fibronectin by Human Dermal Fibroblasts

Human dermal fibroblasts were cultured on tissue culture plastic, gelatin scaffolds, and gelatin/galectin-3 scaffolds for 3 and 7 days with antibodies specific to fibronectin utilized to assess matrix formation. Staining of filamentous actin (red) demonstrated that cells were attached and spread at days 3 and 7 post-seeding on all scaffold types comparable to tissue culture plastic (TCP) (Figure 4C). Staining for fibronectin revealed deposition by fibroblasts on both gelatin and gelatin/galectin-3 scaffolds at days 3 and 7 post-seeding. Increased deposition was seen qualitatively on all tested scaffolds and TCP at day 7 versus day 3 samples, although no observable differences in the immunoreactivity for fibronectin were evident between the scaffold types at either time point.

### 3.7. Gelatin/Galectin-3 Scaffolds Do Not Alter Skin Closure Kinetics in Wild-Type Mice

To determine whether gelatin/galectin-3 scaffolds influence wound closure kinetics in WT mice, four experimentally created wounds were treated with a gelatin scaffold, a gelatin scaffold loaded with 6.7 μg/mL galectin-3, topical delivery of 6.7 μg/mL galectin-3, or no treatment (left empty). Wound closure rates were compared between wounds on day 5 and day 7 following surgery (Figure 5A,B). On both days 5 and 7, no significant differences in closure rate were observed between any treatment conditions) (*p* > 0.05, *N* = 6). Assessment of re-epithelialization showed no differences in wound bed coverage (Figure 5C,D) (*p* > 0.05, *N* = 6) or in epithelial thickness (Figure 5E,F) (*p* > 0.05, *N* = 6) at either time point. The assessment of epithelial migration using antibodies for keratin 17 showed no difference in overall migration distance, but confirmed expression of the pro-migratory keratin (Figure 6). The assessment of tissue structure using Masson’s trichrome staining at day 5 and day 7 post-wounding showed that a similar tissue structure was evident in all wound treatments, with a visible eschar still present over all wounds (Figure 7). At day 7, qualitative increases in collagen were evident in wounds treated with topical galectin-3 (6.7 µg/mL) treated compared to all other treatments (Figure 8). All wounds for each condition were completely re-epithelialized by day 17 post-surgery.

### 3.8. The Influence of Topical Galectin-3 and Gelatin/Galectin-3 Scaffolds on Macrophage Populations during Excisional Healing

To assess if gelatin/galectin-3 scaffolds or topical galectin-3 influenced inflammatory cell infiltration, tissue was labeled using antibodies specific for iNOS (M1 macrophages) and arginase-I (M2 marker) (Figure 9). By day 5, arginase-I positive macrophages were the dominant macrophage population in all tested conditions, localizing to the developing granulation tissue (Figure 9A). Arginase-I-positive macrophages were evident directly under the migrating epithelium and eschar in all conditions tested. Arginase I-positive macrophages were quantified across four treatment groups at day 5 post-wounding. More arginase I-positive macrophages were present in the wounds treated with topical galectin-3 and gelatin scaffolds compared to untreated galectin-3/gelatin scaffolds (*N* = 3 mice, *n* = 3 sections, *p* < 0.05) (Figure 9B). Topical addition of galectin-3 increased arginase-I positive macrophages in comparison to galectin-3/gelatin scaffolds (*N* = 3 mice, *n* = 3 sections, *p* < 0.05) (Figure 9B). 

### 3.9. Increasing the Topical Galectin-3 Concentration Does Not Influence the Wound Closure Rate, Epithelial Structure, or Arginase-I Population Density up to Day 9 Post-Wounding

As topical galectin-3 showed the potential to increase M2 macrophage number during healing, we next assessed whether the concentration of Galectin-3 impacted the wound closure rate or macrophage M2 polarization. The assessment of wound closure kinetics demonstrated no difference in closure rate between any of the tested conditions (Figure 10A,B), with time being the most significant source of variation in closure rate (Figure 10C). On days 7 and 9, a similar structure was evident in all conditions, as assessed using Masson’s trichrome (Figure 11A). Analysis of the epithelial tongue length showed that addition of topical galectin-3 [12.5 µg/mL] significantly increased the length of the tongue compared to empty wounds and topical galectin-3 [25 µg/mL] at day 7 (Figure 11B, *p* < 0.05). No difference was seen in epithelial thickness at day 7 (Figure 11C). Furthermore, by day 9, no differences were evident in epithelial tongue length (Figure 11D) or epithelial thickness (Figure 11E) between the tested conditions. The assessment of collagen content in granulation tissue at day 7 post-wounding using picrosirius red and polarized light demonstrated that the addition of galectin-3 increased collagen birefringence, although no differences were evident based on galectin-3 concentration (Figure 12). IHC labeling of tissue at day 9 for arginase I showed similar levels of positive cells in all tested conditions (Figure 13A,B, *p* > 0.05). 

## 4. Discussion

Chronic skin wounds are problematic as they persist in a pro-inflammatory state and are unable to progress to the proliferative phase of healing and restoration of the barrier [5]. Galectin-3 is a protein implicated in monocyte migration [12], alternative macrophage M2 activation [14], collagen production [11,37], and re-epithelialization [38,39]. We hypothesized that the local delivery of galectin-3 could enhance wound closure by regulating inflammation and increasing re-epithelialization. We first investigated an electrospun fibrous scaffold structure to deliver galectin-3 as it provides a large surface area for the distribution of the protein, for cell adhesion and migration, and to protect it from biodegradation [40,41,42]. Secondarily, we also compared the topical delivery of exogenous galectin-3 to assess whether it exhibits potential as an adjunct therapy to enhance wound closure. 

Electrospinning has received much attention because the structure mimics the size of native extracellular matrix fibers normally present in granulation tissue [43]. For this current study, electrospinning was utilized to produce the scaffolds, with type B bovine gelatin used as the main structural element for the fibers. Electrospinning is a useful method to produce scaffolds with a similar structure to developing granulation tissue [44]. We selected gelatin as it is derived from collagen [45], which represents the primary structural protein of the dermal extracellular matrix [46]. Thus, gelatin provides some chemical similarity to the extracellular matrix while reducing the cost of scaffold fabrication [47]. The use of 20% weight gelatin produced detectable amounts of beaded fibers, which are thought to negatively impact cellular interactions with scaffolds [26], so optimized scaffolds with 21% weight gelatin that eliminated the fibrous beads were fabricated while retaining a suitable fiber diameter within the range of collagen fibril sizes found in human tissues [48]. The porosity obtained in the optimized scaffolds fell within the range of porosities shown in scaffolds electrospun using a variety of polymers and electrospinning parameters, ranging from approximately 60 to 90% [49,50]. Although the obtained porosity is sufficient, obtaining a porosity of 90% has been suggested to be optimal [43]. The electrospun scaffolds used in this study had a low average pore diameter (pore size) of 1.15 μm [51]. Increasing the density of pores in the 50 of 100 μm range would have also been preferable to coincide with the size of the cells on the scaffold and support their infiltration. In fact, scaffolds having pore sizes of approximately 100 μm and porosity in the 90% range have been shown to support the infiltration of cells from the surface of the scaffold [52]. However, it is technically difficult to obtain scaffolds with both pore sizes in this range and fiber diameters in the 100 nm range as decreasing fiber diameter is associated with decreasing pore sizes [53]. As a result, many approaches for scaffold fabrication have also been utilized [54]. In recent years, the 3D printing of scaffolds has gained traction [55], but the clinical environment remains dominated by both natural and synthetic membranes [56], decellularized extracellular matrix scaffolds [57], and skin substitutes [58].

Recombinant human galectin-3 was added to the electrospinning solution during scaffold fabrication to a final concentration of 6.7 μg/mL. This concentration was chosen as it was within the range used by other groups to achieve effects in vitro. For example, in studies in the skin, concentrations as low as 1μg/mL have been used to increase keratinocyte migration speed [19] and galectin-3 has previously been shown to have a concentration-dependent effect on monocyte recruitment from 0.001 to 0.01 μM [16]. Additionally, it has been demonstrated that 6.3 μg/mL promoted human keratinocyte migration, while higher concentrations (50 μg/mL) inhibited migration in vitro [59]. The detection of galectin-3 protein sequences from scaffold samples confirmed that the blend electrospinning method could generate scaffolds with galectin-3 dispersed throughout the fibers. The sequences identified are located within the protein’s CRD, which is important as this domain is required for many of the protein functions [13,14,38,60]. The identification of galectin-3 was expected, as several groups have previously used the blend electrospinning method for the delivery of matricellular proteins and growth factors [31,32,61]. 

During healing, the granulation tissue is essential in guiding cells into the wound by providing a matrix for supporting cell adhesion and migration [9,62]. The ability of cells to adhere and proliferate on the scaffolds was therefore used as a measure of biocompatibility, although not bioactivity. Using human dermal fibroblasts, we show that the initial adhesion of human dermal fibroblasts was increased relative to the tissue culture plastic by both gelatin and gelatin/galectin-3 scaffolds. This increased adhesion in both the gelatin and gelatin/galectin-3 scaffolds likely results from the arginine–glycine–aspartate (RGD) sequences contained within gelatin, which is known to promote cell adhesion through integrin binding [63]. The improved adhesion in both scaffolds can also be attributed in part to the increased surface area that the scaffolds offer for attachment, further promoting cell-matrix interactions [64]. Over a two-week period, the proliferation profile of human dermal fibroblasts on the gelatin and gelatin/galectin-3 scaffolds was consistent with that of the tissue culture plastic controls. This finding confirms that both the gelatin and gelatin/galectin-3 scaffolds were non-toxic and able to support cell growth. Other groups have shown that the proliferation of human dermal fibroblasts is similar on gelatin scaffolds compared to cells grown on tissue culture plastic controls [27,65]. Proliferation on scaffolds that is equal to or higher than culture on tissue culture plastic was an important finding, as scaffolds made from other materials, including chitosan and polycaprolactone, have demonstrated reduced rates of proliferation [44,65,66]. Both gelatin and gelatin/galectin-3 scaffolds also supported the deposition of fibronectin by fibroblasts with the secretion of this protein important during wound healing where it mediates cell adhesion and migration, stimulates collagen deposition, and contributes to wound contraction [67]. Our results are consistent with previous studies, showing that fibronectin increases in dermal fibroblasts in response to exogenous galectin-3 [68]. 

As the scaffolds demonstrated biocompatibility in vitro, we next assessed the influence of each scaffold as well as topical delivery of galectin-3 on wound closure kinetics in a C57BL/6 mouse model of acute excisional wound healing. The use of gelatin scaffolds with and without the addition of galectin-3 did not significantly alter the wound closure kinetics in WT mice over a 17-day period. No evidence of a foreign body response was seen histologically [69]. The addition of topical galectin-3 similarly had no influence on wound closure kinetics, but this does not eliminate the possibility of changes early in the inflammatory phase that do not manifest in measurable closure changes, which arise primarily due to the contraction of the wounds [31,32]. Although treating wounds with topical galectin-3 and scaffolds containing galectin-3 did not significantly increase wound closure kinetics, this is supported by our previous finding that wound closure kinetics are not impaired in galectin-3 knockout mice [20]. Interestingly, gelatin scaffolds alone have previously been reported to increase wound closure in a full-thickness skin model in rats, which is in part supported by our findings, with gelatin scaffolds showing more efficacy than gelatin/galectin-3 scaffolds [49]. We have previously shown that both periostin and connective tissue growth factor 2 (CCN2) remain biologically active and enhance wound repair after electrospinning [31,32], suggesting that it is likely the conformation of galectin-3 itself rather than electrospinning per se that reduces the effectiveness of wound closure. As with any study, we acknowledge limitations. It is possible, particularly in the electrospun scaffolds, that the galectin-3 does not have the necessary structure for cell binding and could exhibit reduced activity. Potential conformational changes in the protein, even when added exogenously, could explain the variation in results from the different studies examining galectin-3 in excisional skin repair. Future work should focus on the structure of galectin-3, delivery mechanisms, and how this corresponds to biological effects on monocytes, fibroblasts, and keratinocytes individually. As galectin-3 is a ligand for many different molecules [9,70], it is also conceivable that it could be sequestered and unavailable to regulate specific aspects of skin repair. Future studies should focus on higher concentrations, which could be more effective in eliciting cell responses evident in vitro. 

In wound healing, inflammation follows hemostasis, a process during which monocytes are recruited to the wound by chemoattractants and differentiate into macrophages [18]. Macrophages are vital constituents of the wound-healing process, mediating wound healing through the release of regulatory molecules, which are based on their phenotype [71]. Classically activated (M1) macrophages produce nitric oxide and secrete pro-inflammatory cytokines including TNF-α, IL-1, IL-6, and IL-12, while alternatively activated macrophages (M2) are implicated in tissue remodeling and secrete TGF-β [72]. Galectin-3 has previously been implicated in macrophage function [13,14,16]; therefore, macrophage populations were investigated after treatment with topical galectin-3 and gelatin/galectin-3 scaffolds in order to discern whether exogenous human recombinant galectin-3 could increase the number of M2 polarized macrophages. At day 7, topical galectin-3 [6.7 µg/mL] significantly increased the number of arginase-I positive cells compared to gelatin/galectin-3 scaffolds and untreated wounds. Mackinnon et al. reported that bone marrow-derived macrophages (BMDMs) from galectin-3 knockout mice show a defect in IL-4 and IL-13 M2 macrophage polarization in vivo and in vitro [14]. However, this study did not test the addition of exogenous galectin-3 on macrophage polarization; therefore, there is currently no indication as to whether its use would be sufficient in rescuing the deficient M2 polarization of BMDMs in galectin-3 knockout mice. Future work should assess the influence of exogenous galectin-3 in upregulating the expression of surface-bound galectin-3, as well as the secretion of galectin-3 or upregulating CD98, which are each implicated in the suggested autocrine loop that controls M2 activation [14]. 

During wound healing, keratinocyte proliferation and migration occur to restore the epithelial barrier [62]. Previous studies of excisional healing using galectin-3 knockout mice have shown a decrease in the length of the epithelial tongue and decreased re-epithelialization at days 2 [19] and 7 post-wounding [20]. This deficient re-epithelialization arises due to a migratory defect in keratinocytes caused by aberrant epidermal growth factor receptor (EGFR) endocytosis and recycling. Indeed, the presence of cytosolic galectin-3 was shown to mediate EGFR receptor recycling through binding to the cytoplasmic protein ALG-2-interacting protein X (ALIX) [19]. When recombinant human galectin-3 [6.7 µg/mL] was added to wounds of WT mice topically or using a gelatin scaffold, differences in epithelial thickness were not quantified at days 5 or 7 post wounding. This result was consistent with previous reports showing no defect in epithelial thickness in galectin-3 knockout mice [19,20]. Differences in re-epithelialization were also not observed in both WT mice at day 5 and 7 following wounding, which was consistent with the finding that exogenous galectin-3 was not effective in correcting the defective EGFR endocytosis and recycling in galectin-3 knockout mice [19]. However, by changing the galectin-3 concentration to 12.5 µg/mL, but not 25 µg/mL, in our pilot study, an increase in epithelial tongue length was observed, suggesting that the concentration of galectin-3 may exert effects on keratinocytes. Interestingly, studies focused on corneal healing have shown that exogenous human recombinant galectin-3 can increase re-epithelialization in WT mice [38] and in monkey corneal explants [39]. This increase was suggested to be an indirect modulation of galectin-7 expression by the presence of exogenous galectin-3. Galectin-7 has been shown to accelerate re-epithelialization in galectin-3 knockout mice and of importance, mouse embryonic fibroblasts from galectin-3 knockout mice showed reduced levels of galectin-7 [38]. Of significance for skin healing, it has been shown that the expression of galectin-7 is not altered at day 7 following wounding in WT mice, implying that the mechanism described in the cornea might not be mimicked completely in skin healing [20]. This discrepancy highlights the general issue of context-specific roles of matricellular proteins [9].

In our study, using picrosirius red staining, we demonstrate that delivery of galectin-3 in an electrospun form as well as in saline results in increased collagen content in the granulation tissue. Our findings agree with previous studies that show that in wounds treated with 20 ng/mL galectin-3, the collagen score was found to be elevated in excisional wound repair in rats, which was concomitant with the increased tensile strength of the skin [68]. The expression of COL1A1, COL1A2, and COL3A1 has been shown to increase in vitro in human dermal fibroblasts, and pharmacological inhibition of galectin-3 results in reduced cardiac fibrosis [73]. However, galectin-3 expression in granulation tissue is downregulated by day 7 in 10 mm full-thickness excisional wounds in rats [24]. As the peak of the proliferative phase is day 7 [31], it is possible that galectin-3 modulates fibroblast collagen synthesis as the cells are actively migrating into the granulation tissue; we did identify increased collagen birefringence under the intact skin when galectin-3 was delivered. A role for galectin-3 in the myofibroblast transition of dermal fibroblasts has been described, although, in the same study, galectin-3 was found to have no significant effect on fibronectin synthesis [23], findings that we confirmed in this study. In contrast to our study and that of Gal et al. [68], Dvoránková and colleagues were unable to identify any effect of topical galectin-3 on wound healing in vivo [23]. However, by focusing on re-epithelialization and wound contraction specifically, no measure was made of collagen content. Based on our findings and those of Gal and colleagues, as well as the defined role of galectin-3 in fibrosis [11], we conclude that local delivery of the lectin could be used to augment collagen synthesis in situations of impaired healing. 

## 5. Conclusions

In summary, electrospun gelatin/galectin-3 scaffolds were developed, which show biocompatibility when tested both in vitro and in vivo. Gelatin/galectin-3 scaffolds exhibited no overall effect on the closure of excisional wounds in C57BL/6 mice compared to other treatments, although it increased collagen synthesis in the granulation tissue. Interestingly, topical galectin-3 showed potential for increasing M2 macrophage number, enhancing epithelial migration, and increasing collagen production, although it was specific to the concentration of Galectin-3 in the solution. A limitation of the study was that we only assessed the delivery of galectin-3 using gelatin or in a topical solution, and it is possible that more appropriate mechanisms of delivery could be developed. Moreover, additional time points and the assessment of epithelial differentiation could provide additional information on the influence of galectin-3 on keratinocyte behavior in vivo. A more detailed assessment of the tensile strength of skin in the presence of galectin-3 would also provide additional information on the role of the lectin in repair mechanisms. Future studies should focus on the delivery of galectin-3 in delayed healing models such as the db/db murine model of type II diabetes, which shows impaired re-epithelialization. We conclude that the efficacy of galectin-3 on skin healing is likely dependent on the mode of delivery and conformational availability to infiltrating cells. 

## Figures and Tables

**Figure 1 bioengineering-11-00960-f001:**
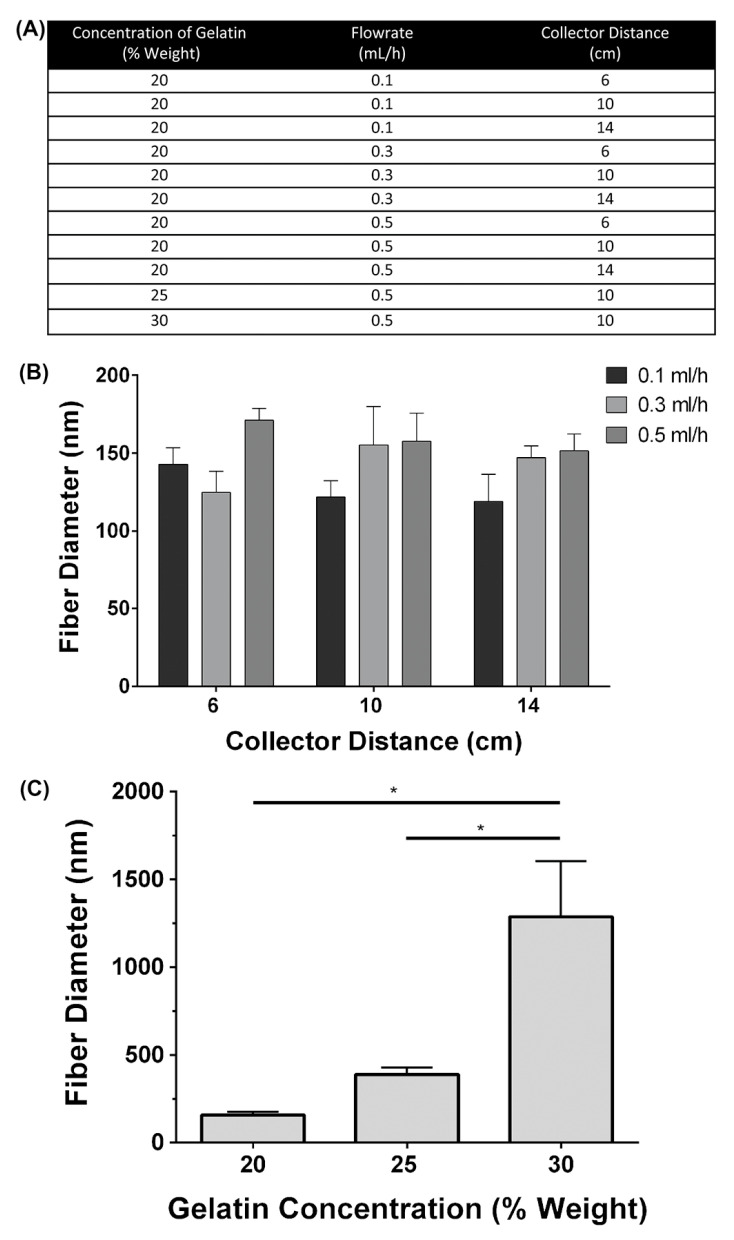
Influence of electrospinning parameters on fiber size (**A**). A summary of parameter combinations used during electrospinning to compare the effects of the flowrate and collector distance on the resulting fiber diameter. (**B**) Effect of increasing collector distance and flowrate on mean fiber diameter. The flowrate was varied between 0.1, 0.3, and 0.5 mL/h and the collector distance was varied between 6, 10, and 14 cm. The concentration of gelatin was held constant at 20% weight. No significant differences in fiber diameter were observed at any of the conditions assessed. *N* = 3, *n* = 250, two-way ANOVA, *p* > 0.05. All data are represented as mean ± SEM. (**C**) Mean fiber diameters measured at 20%, 25%, and 30% weight gelatin. The flowrate and collector distance were held constant at 0.5 mL/h and 10 cm, respectively. Fibers electrospun at a concentration of 30% weight gelatin had significantly higher fiber diameters than those electrospun at 20 or 25% (*N* = 3, *n* = 250, two-way ANOVA, * = *p* < 0.05).

**Figure 2 bioengineering-11-00960-f002:**
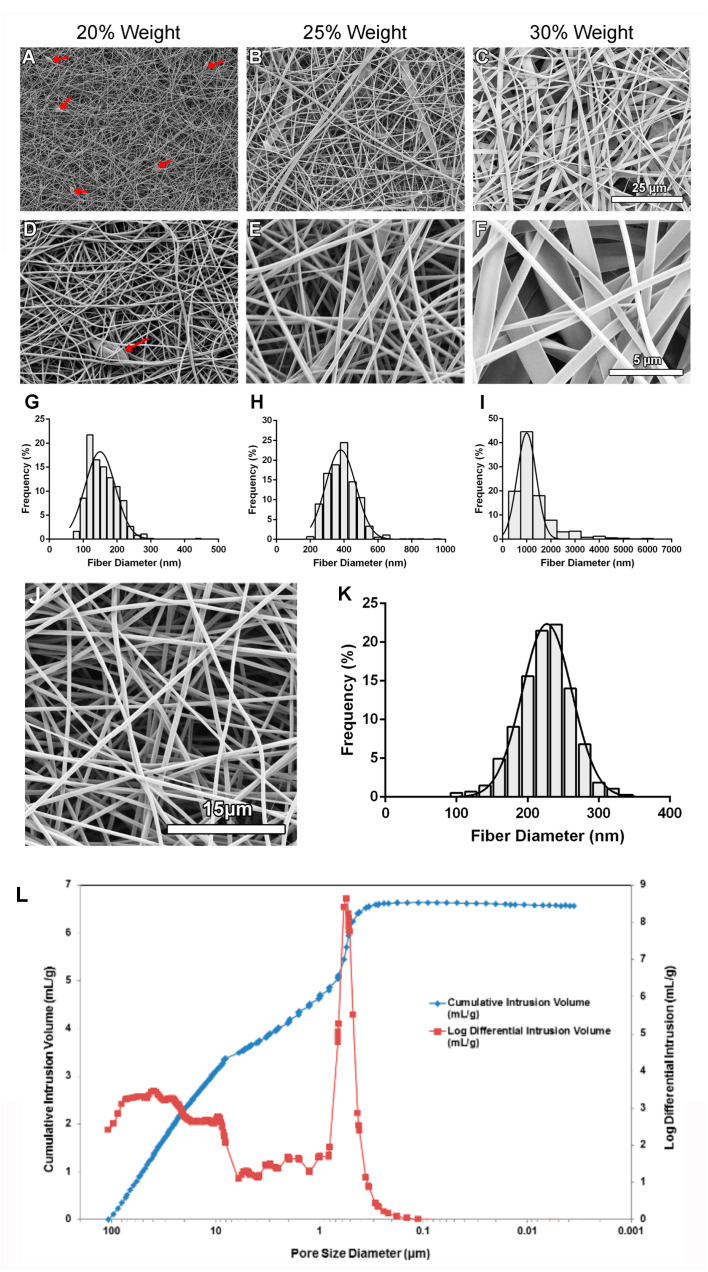
The effect of increasing the gelatin concentration on fiber morphology and fiber diameter distribution. (**A**–**F**): Images of electrospun gelatin fibers using scanning electron microscopy. Fiber mats were electrospun using a flowrate of 0.5 mL/h, collector distance of 10 cm, and varying gelatin concentrations. (**A**–**C**): Images were taken at 1000× magnification. (**D**–**F**): Images were taken at 5000× magnification. Red arrows indicate beading. (**G**–**I**): Frequency distribution graph showing the percentage of fiber diameters (from 750 measurements across *N* = 3 independent experiments) within each bin range for scaffolds electrospun using a flowrate of 0.5 mL/h, a collector distance of 10 cm, and a gelatin concentration of 20%, 25%, and 30% weight. *N* = 3, *n* = 250. (**G**): Bin size: 20 nm. (**H**): Bin size: 50 nm. (**I**) Bin size: 500 nm. (**J**) SEM image of a scaffold electrospun using a flowrate of 0.5 mL/h, a collector distance of 10 cm, and a gelatin concentration of 21% weight. (**K**) Frequency distribution graph showing the percentage of fiber diameters within each bin range for scaffolds electrospun using a flowrate of 0.5 mL/h, a collector distance of 10 cm, and a gelatin concentration of 21% weight. *N* = 3 independent experiments, *n* = 250. Bin size: 20 nm. (**L**) Mercury porosimetry pore size distribution plot. Representative graph of pore diameter distribution measured as a function of differential and cumulative intrusion volumes. The cumulative pore volume curve shows steeper slopes between 10 and 100 μm and 0.1 and 1 μm, coinciding with peaks in the log differential intrusion volume. Each log differential intrusion value represents the relative quantity of mercury entering pores of a specific size. Mercury porosimetry was repeated three times (*N* = 3) on multiple different batches of scaffolds.

**Figure 3 bioengineering-11-00960-f003:**
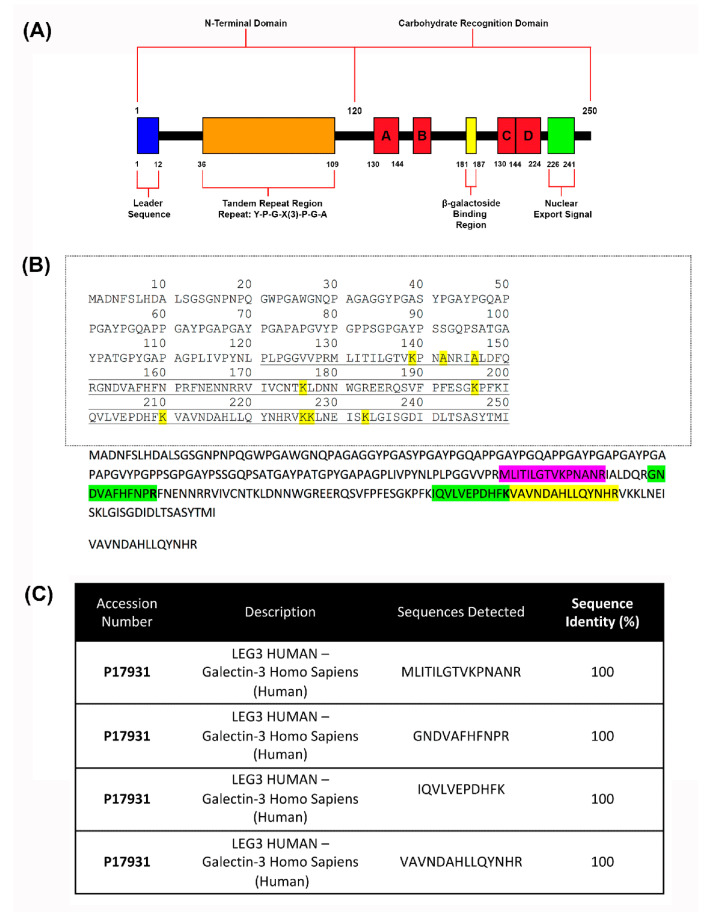
Detection of the galectin-3 protein after scaffold fabrication using mass spectrometry. (**A**) Domains and structures of recombinant human galectin-3. It features a 120 amino acid N-terminal region that contains a leader sequence and a tandem repeat region rich in proline, glycine, and arginine. It also comprises a CRD containing a β-galactoside binding region and a sequence required for nuclear export. (**B**) Visualization of detected sequences on recombinant human galectin-3: Mass spectrometry was conducted three times for the detection of galectin-3. Four peptide sequences with 100% alignment with the human recombinant galectin-3 protein (UniProt Accession # P17931) structure in the carbohydrate recognition domain were detected. (**C**) The 15 amino acid sequence of MLITILGTVKPNANR aligns with the protein at amino acid locations 130–144. The 11 amino acid sequence of GNDVAFHFNPR aligns with the protein at amino acid locations 152–162. The 11 amino acid sequence of IQVLVEPDHFK aligns with the protein at amino acid locations 200–210. The 14 amino acid sequence of VAVNDAHLLQYNHR aligns with the protein at amino acid locations 211–225. Green, yellow and pink show location of detected sequences.

**Figure 4 bioengineering-11-00960-f004:**
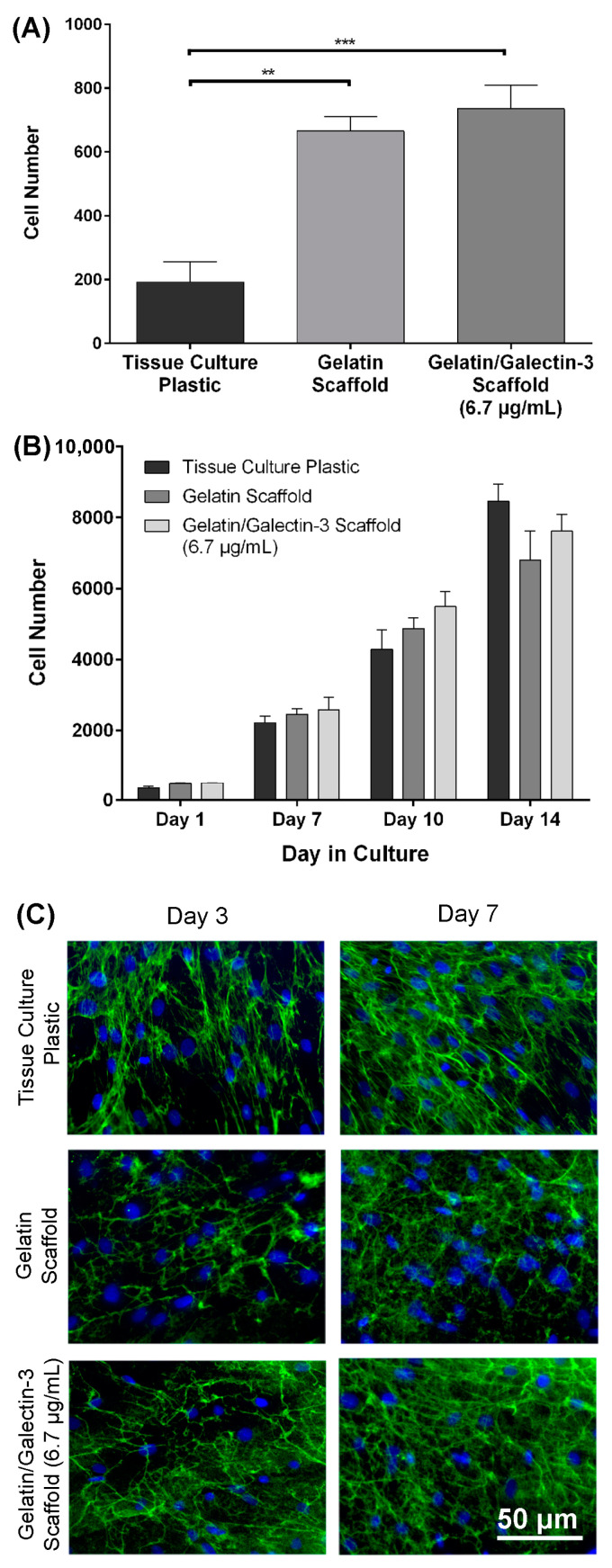
Biocompatibility testing of gelatin/galectin-3 scaffolds using human dermal fibroblasts. (**A**) Human dermal fibroblasts were seeded onto tissue culture plastic, gelatin scaffolds, and gelatin/galectin-3 scaffolds (6.7 μg/mL) and left to attach for one hour. Cell numbers were significantly higher in wells containing the gelatin scaffolds and gelatin/galectin-3 scaffolds than in tissue culture plastic wells. *N* = 3 independent experiments, *n* = 3 repeats per condition, one-way ANOVA, Tukey post-test for multiple comparisons, ** *p* < 0.01, *** *p* < 0.001. All data are represented as mean ± SEM. (**B**) Human dermal fibroblasts were cultured on tissue culture plastic, gelatin scaffolds, and gelatin/galectin-3 scaffolds (6.7 μg/mL) over 14 days. There were no significant differences in the cell number between the groups at all time points assessed. *N* = 3 independent experiments, *n* = 4 repeats per condition, two-way ANOVA, *p* > 0.05. All data are represented as the mean ± SEM. (**C**) Deposition of fibronectin by human dermal fibroblasts on scaffolds. Human dermal fibroblasts were cultured on tissue culture plastic, gelatin, and gelatin/galectin-3 scaffolds (6.7 μg/mL) over 7 days. Representative images show fibronectin (green) and cell nuclei (blue) using immunocytochemistry. There were no observable differences in the amount of fibronectin deposited between the scaffolds. *N* = 3 independent experiments, *n* = 3 repeats per condition.

**Figure 5 bioengineering-11-00960-f005:**
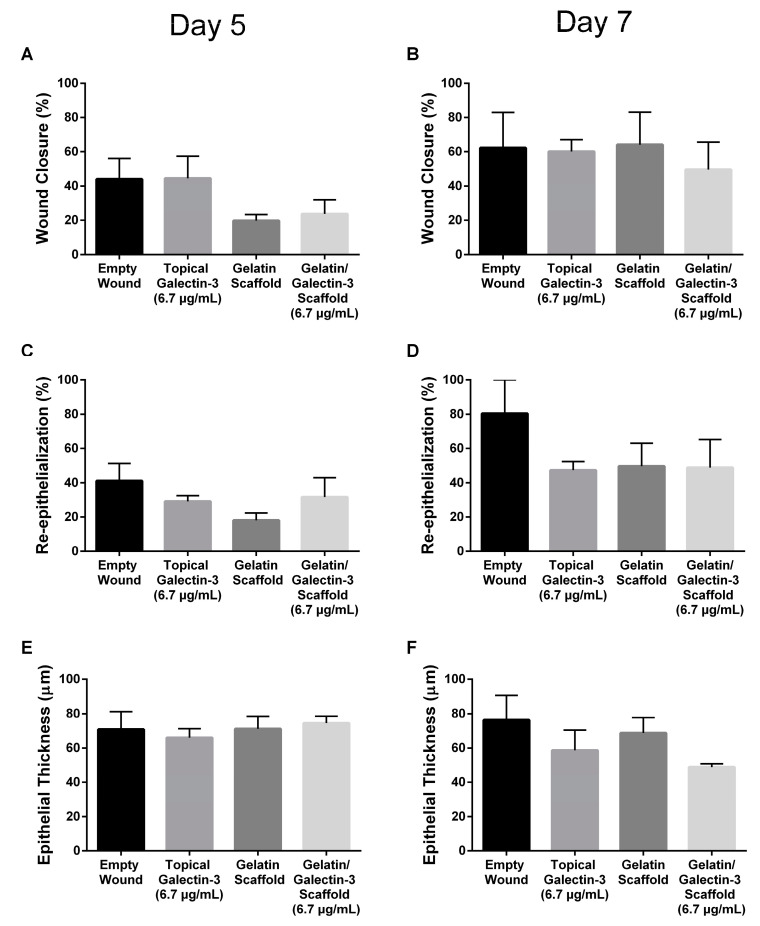
Wound closure, re-epithelialization, and epithelial thickness in WT mice. Full-thickness excisional wounds measuring 6 mm in diameter were treated with topical galectin-3 (6.7 μg/mL), a gelatin scaffold, a gelatin/galectin-3 scaffold (6.7 μg/mL), or left empty (control). (**A**,**B**) The percentage of closure relative to the original wound was calculated on day 5 (*N* = 3) and day 7 (*N* = 3). There were no significant differences in closure between each of the treatment groups on days 5 and 7. *N* = 3, one-way ANOVA, Tukey post-test for multiple comparisons, *p* > 0.05. (**C**,**D**) The percentage of re-epithelialization was calculated on day 5 (*N* = 3) and day 7 (*N* = 3). There were no significant differences in re-epithelialization between each of the treatment groups on days 5 and 7. *N* = 3, one-way ANOVA, Tukey post-test for multiple comparisons, *p* > 0.05. (**E**,**F**) The thickness of the epithelium was calculated on days 5 (*N* = 3) and day 7 (*N* = 3). There were no significant differences in the epithelial thickness between each of the treatment groups on days 5 and 7 following wounding. *N* = 3, one-way ANOVA, Tukey post-test for multiple comparisons, *p* > 0.05. All data are represented as mean ± SEM.

**Figure 6 bioengineering-11-00960-f006:**
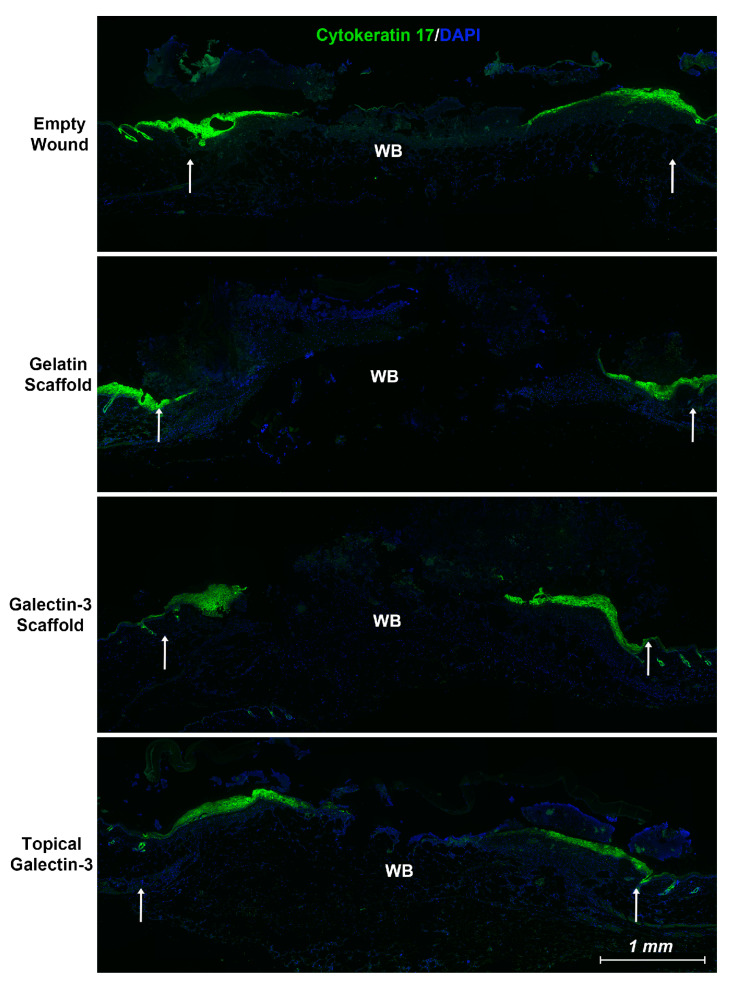
Local delivery of galectin-3 does not increase the distance of re-epithelialization significantly. Using an antibody specific for the keratin 17 epitope, we show that the epithelial tongue in all tested conditions is positive for keratin 17, which has been associated with a pro-migratory phenotype in cancer [35,36]. Wounds from day 5 are shown. WB = wound bed. White arrows highlight the wound edge.

**Figure 7 bioengineering-11-00960-f007:**
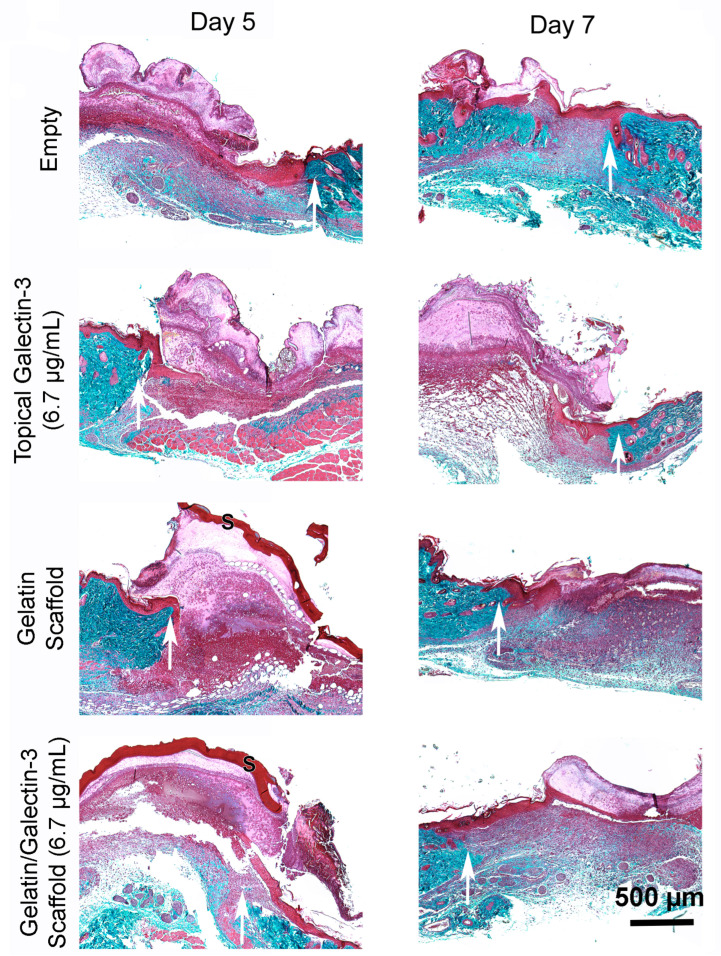
Assessment of tissue structure post wounding on day 5 and day 7. Masson’s Trichrome staining following in vivo full-thickness wounding in WT mice. Full-thickness excisional wounds measuring 6 mm in diameter were treated with topical galectin-3 (6.7 μg/mL), a gelatin scaffold, a gelatin/galectin-3 scaffold (6.7 μg/mL), or left untreated (control). Sections show the wound edge for each condition and the epithelial tongue. For consistency, images shown of the four treatment conditions are from the same mouse on day 5 and day 7. White arrows show the wound edge. (*N* = 3 for each time point).

**Figure 8 bioengineering-11-00960-f008:**
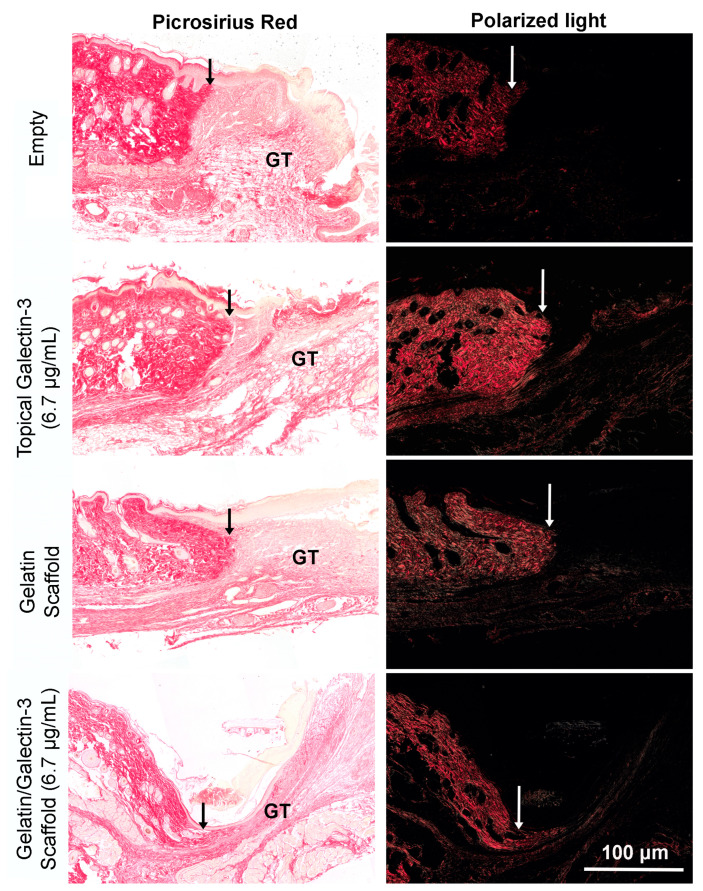
Assessment of collagen content in granulation tissue at day 7 post-wounding using picrosirius red and polarized light. Full-thickness excisional wounds measuring 6 mm in diameter were treated with topical galectin-3 (6.7 μg/mL), a gelatin scaffold, a gelatin/galectin-3 scaffold (6.7 μg/mL), or left untreated (control). Sections show the wound edge for each condition and the maturing granulation tissue. For consistency, the representative images shown of the four treatment conditions are from the same mouse on day 7. White and black arrows show the wound edges. GT = granulation tissue. (*N = 3*).

**Figure 9 bioengineering-11-00960-f009:**
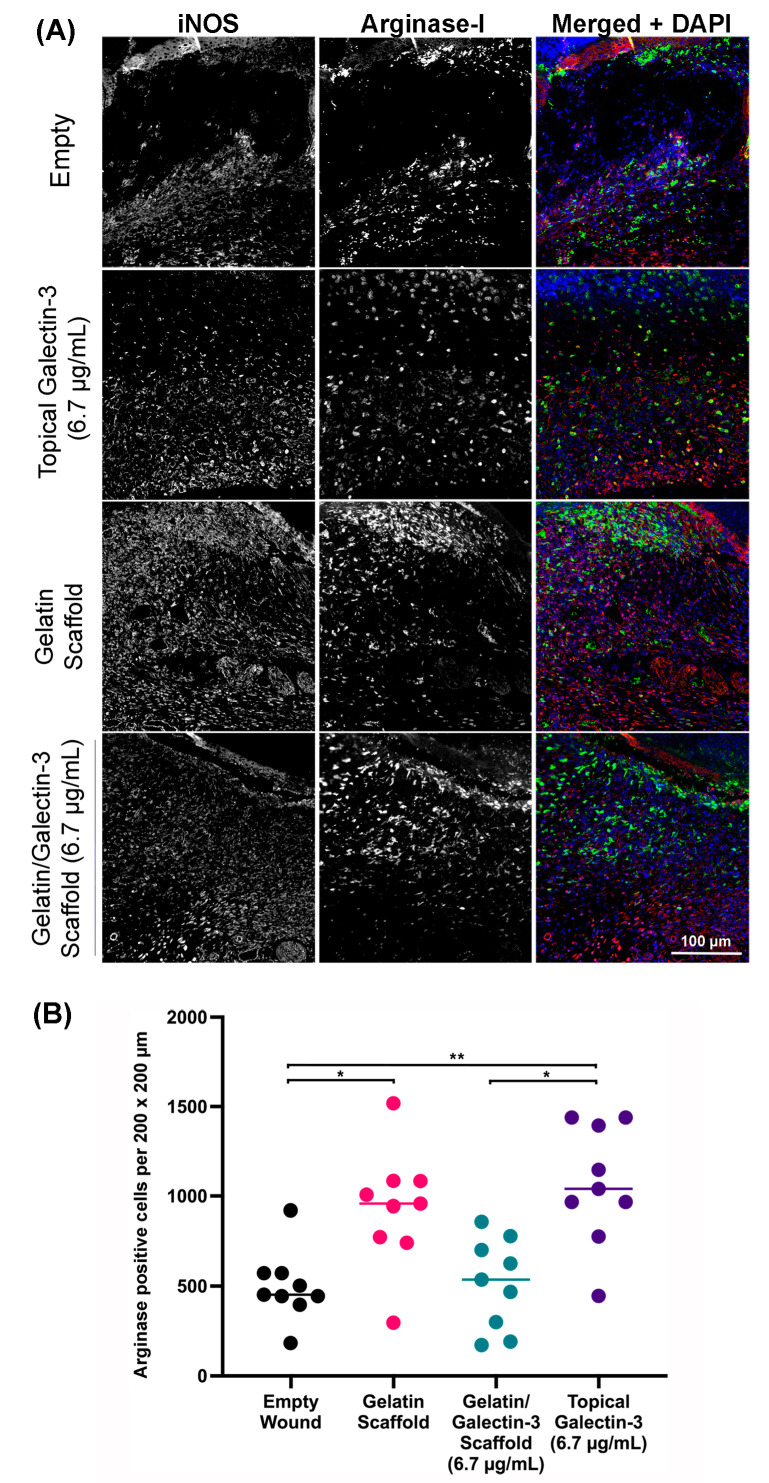
Quantification of arginase-I macrophage populations at day 5 post wounding. (**A**) Sections show the relative amounts of arginase I-positive macrophages (green) and iNOS-positive macrophages (red) in the wound bed for each treatment condition. Cell nuclei are shown in blue. Images shown are representative of sections collected from three separate mice (*N* = 3). (**B**) The density of arginase I-positive macrophages within the wound was determined in WT mice at day 5 following wounding. *N* = 3 mice, *n* = 3 sections per wound, one-way ANOVA, Tukey post-test for multiple comparisons, * = *p* < 0.05, ** = *p* < 0.01. All data are represented as mean ± SD.

**Figure 10 bioengineering-11-00960-f010:**
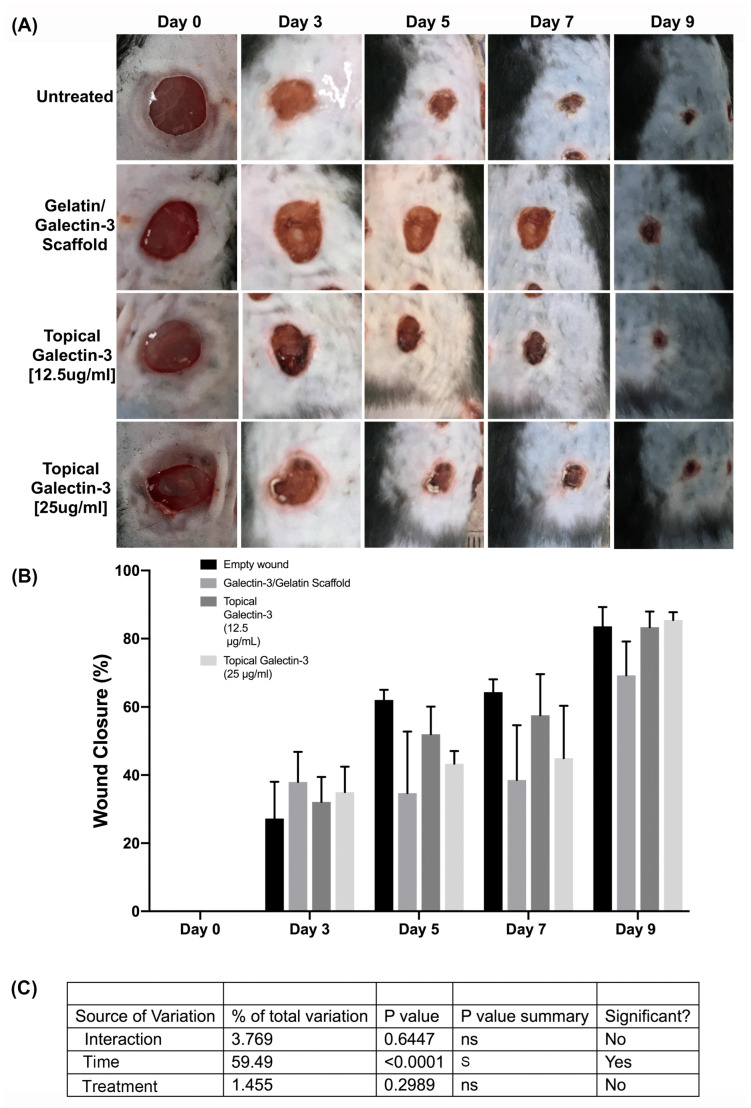
Preliminary assessment of the galectin-3 concentration on wound closure kinetics in full-thickness wounds. (**A**) Full-thickness excisional wounds measuring 6 mm in diameter were treated in an electrospun scaffold composed of 21% gelatin blended with galectin-3 (6.7 μg/mL), daily topical treatment of galectin-3 in PBS (12.5 μg/mL or 25 μg/mL), or left empty (no scaffold control and daily topical PBS). Representative images of the four conditions are shown at days 0, 3, 5, 7, and 9 post-wounding. (**B**) Percentage of wound area closure was calculated over a 9-day period. ANOVA, *p* > 0.05 between each treatment and the control scaffold. Data are represented as mean ± SEM. (**C**) Interaction analysis demonstrated that time post-wounding was the significant source of variation (Row factor).

**Figure 11 bioengineering-11-00960-f011:**
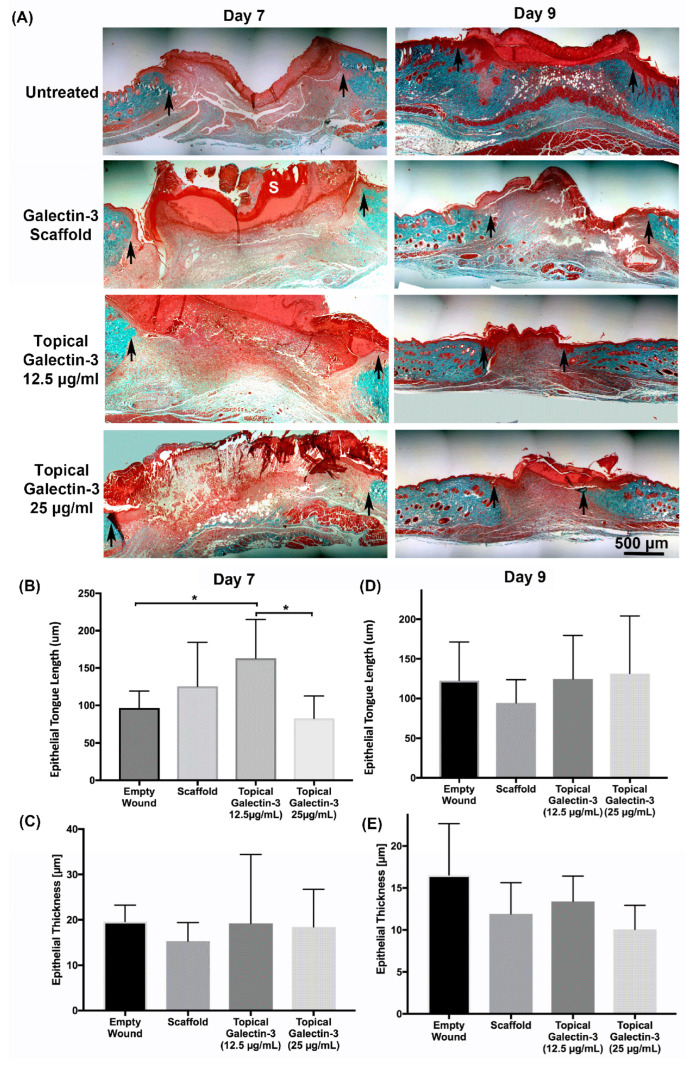
Assessment of tissue structure post wounding. Full-thickness excisional wounds measuring 6 mm in diameter were treated with implantation of an electrospun scaffold composed of 21% gelatin blended with galectin-3 (50 μg/mL), daily topical treatment of galectin-3 in PBS (12.5 μg/mL or 25 μg/mL), or left empty (no scaffold control, daily topical PBS). (**A**) Sections from day 7 and day 9 post-wounding. Quantification of Epithelial tongue length and thickness. Following in vivo full-thickness wounding at (**B**,**C**) day 7 and (**D**,**E**) day 9. Topical galectin-3 [12.5 µg/mL] significantly increased epithelial tongue length versus untreated and topical galectin-3 [25 µg/mL] at day 7. *N* = 2, *n* = 5, one-way ANOVA with Bonferroni post-hoc testing, * = *p* < 0.05. All data are represented as mean ± SEM.

**Figure 12 bioengineering-11-00960-f012:**
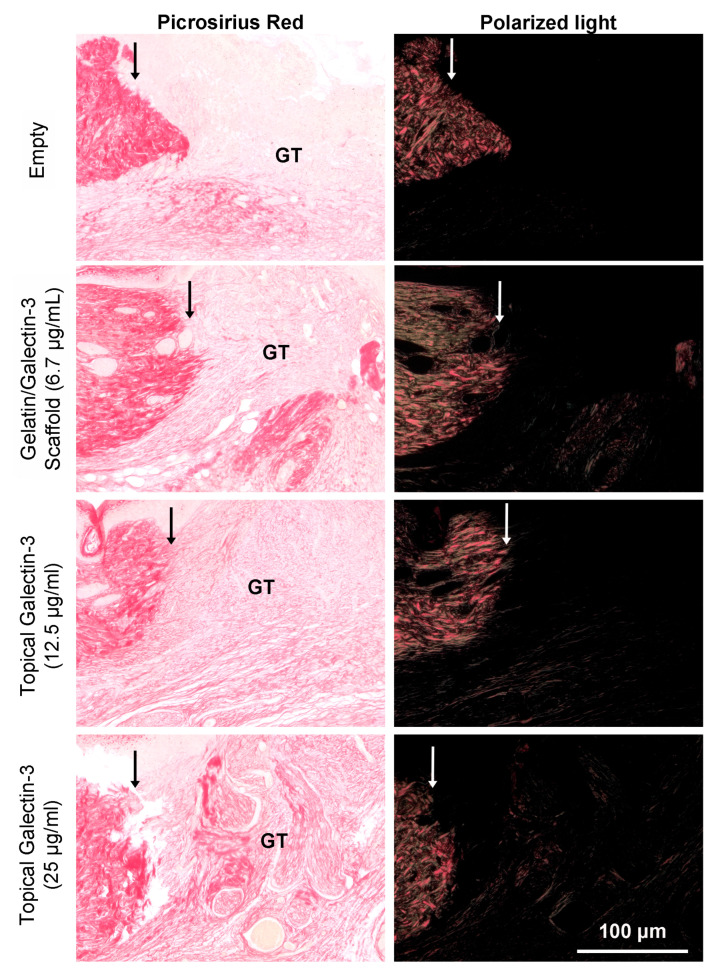
Assessment of collagen content in granulation tissue at day 7 post-wounding using picrosirius red and polarized light. Full-thickness excisional wounds measuring 6 mm in diameter were treated with a gelatin/galectin-3 scaffold (6.7 μg/mL), topical galectin-3 (12.5 μg/mL), topical galectin-3 (25 μg/mL), or left untreated (control). Sections show the wound edge for each condition and the maturing granulation tissue. For consistency, representative images shown of the four treatment conditions are from the same mouse on day 7. White and black arrows show wound edges. GT = granulation tissue.

**Figure 13 bioengineering-11-00960-f013:**
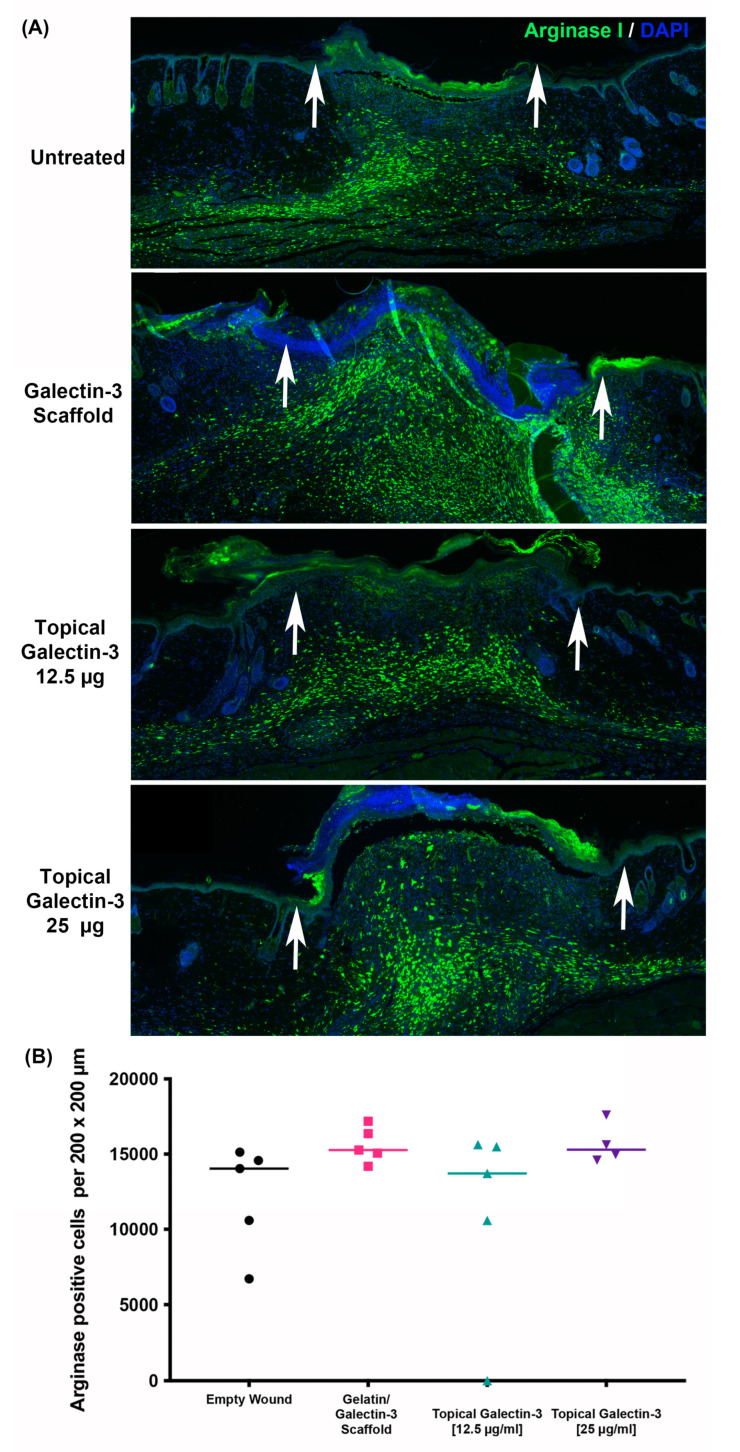
Quantification of Arginase I-Positive Macrophages in Full-Thickness Wounds. (**A**) Full-thickness excisional wounds measuring 6 mm in diameter were treated with implantation of an electrospun scaffold composed of 21% gelatin blended with galectin-3 (6.7 μg/mL), daily topical treatment of galectin-3 in PBS (12.5 μg/mL or 25 μg/mL), or left empty (no scaffold control, daily topical PBS). Sections show the relative amounts of arginase I-positive macrophages (green) in the wound bed at day 7 post-wounding for each treatment condition. Cell nuclei are shown in blue. (**B**) Quantification of Arginase I-Positive Macrophage Density within the Wound Bed. Density of arginase I-positive macrophages was quantified from 5 sections per wound to ensure spatial distributions were assessed within each wound bed. No significant differences in the density of arginase I-positive cells were observed. *N* = 5 mice, *n* = 5 sections per wound per animal, one-way ANOVA with Bonferroni post-hoc testing, *p* > 0.05. All data are represented as mean ± SEM.

## Data Availability

The data presented in this study are available on request from the corresponding author (accurately indicating status).

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
