# Peer review of "Galectin-3/Gelatin Electrospun Scaffolds Modulate Collagen Synthesis in Skin Healing but Do Not Improve Wound Closure Kinetics"

_bioengineering, 2024, doi:10.3390/bioengineering11100960_

Round 1

Reviewer 1 Report (Previous Reviewer 1)

Comments and Suggestions for Authors

1. Why the authors dissolved gelatin in acetic acid? How did they arrive this conclusion? Why can not be other solvents such as NMP, DMF or DMSO used? What is the advantage of using acetic acid? These questions to be addressed under materials and methods section.

2.  generally, nano meter refer to the range between 1 to 100nm. The SEM image of the manuscript displays the size of the thread more than 100nm. The results section the units need to be corrected. 

3. Conclusion should reflect the results, limitation of the work and the future scope. Revise the conclusion accordingly. 

Comments on the Quality of English Language

Spell check and typos to be checked

Author Response

  1. Why the authors dissolved gelatin in acetic acid? How did they arrive this conclusion? Why cannot be other solvents such as NMP, DMF or DMSO used? What is the advantage of using acetic acid? These questions to be addressed under materials and methods section.

We used acetic acid to dissolve the polymers based on prior work. Primarily we used acetic acid to reduce potential cytotoxic effects of other solvents. This has been made clear in the materials and methods section.

  1. Generally, nano meter refer to the range between 1 to 100nm. The SEM image of the manuscript displays the size of the thread more than 100nm. The results section the units need to be corrected. 

While we understand the reviewer’s comment, we are not claiming that these are nanofibers. Respectfully, to refer to something as being 200-300nm in size is reasonable use of scientific notation. While in figure 1 we have measures over 1 µm in size, on the same graph we have measures under 500 nm. Based on the size of the fibres, we respectfully would rather refer to them as 300nm in size than 0.3 µm in size. We do not see this as an unusual preference.

  1. Conclusion should reflect the results, limitation of the work and the future scope. Revise the conclusion accordingly. 

We have reorganized the conclusions of the paper.

Reviewer 2 Report (Previous Reviewer 2)

Comments and Suggestions for Authors

In Figure 4 the authors reported the use of human dermal fibroblast, but these cells are not characterized if they are really dermal fibroblast. Authors must characterized these cells are dermal fibroblast. 

Figure 5 data is not significant at all. 

Figure 6 data is not quantified, additionally data is shown for 5 days only, why not day 7. the figure legend show two references, if this figure is cited from there?

Figure 7 author claims that they have calculated the thickness of epithelium and percentage of re-epithelization but no data is provided. However figure 7 histological images showed in improvement in treated group. 

Figure 8 why data for day is given and why not day 5. There is no quantification and images show no improvement in healing. 

Figure 9B show positive cells per cell cm-2, what this means. How authors analyzed microscopic images in cm. 

Figure 11, 12, and 13 are repeated from figure 7, 8, and 9. No new information are provided. 

There is no novelty in this work, it is repetation of already established fact. 

Comments on the Quality of English Language

Fine

Author Response

1). In Figure 4 the authors reported the use of human dermal fibroblast, but these cells are not characterized if they are really dermal fibroblast. Authors must characterized these cells are dermal fibroblast. 

The cells used in this study were dermal fibroblasts isolated from skin samples harvested by Dr Hamilton’s laboratory. Dr Hamilton’s research group has extensive experience in using primary fibroblasts (Elliott et al, Journal of cell science 125 (1), 121-132, 2012; Kim et al, Journal of Dental Research 92 (11), 1022-1028, 2013; Kim et al, Journal of cellular and molecular medicine 19 (6), 1183-1196, 2015; Walker et al, Journal of Investigative Dermatology 136 (5), 1042-1050 2016; Martin et al Acta Biomaterialia 83, 199-210, 2019; Li et al, Materials 14 (21), 6447, 2019; Kim et al, Scientific Reports 9 (1), 2708, 2019;  Brooks et al, ACS Appl. Mater. Interfaces 2023, 15, 16, 19817–19832) including lineage tracing studies in mice in which cells were comprehensively analyzed for numerous markers (Walker et al, Hamilton, FASEB BioAdvances 3 (7), 541, 2021). Phenotypic modulation of fibroblast populations is one of the core research areas of Dr Hamilton’s research program. We are also slightly confused as to why the reviewer is only highlighting this in review 3. The fibroblasts were used to test biocompatibility of the scaffolds, with the effects of the scaffolds ultimately tested in vivo.

2). Figure 5 data is not significant at all. 

This is a correct interpretation by the reviewer, we are not sure what the reviewer is suggesting about whether something is significant or not with regards to relevance. We had a hypothesis with respect to the use of galectin-3 and we tested that hypothesis using appropriate methods. The findings were that exogenous galectin -3 does not influence these measures during the healing response. Results showing no effect of a treatment are still relevant in science.

3). Figure 6 data is not quantified, additionally data is shown for 5 days only, why not day 7. the figure legend show two references, if this figure is cited from there?

The references cited are simply mentioned with regards to the fact that keratin-17 is associated with a pro-migratory phenotype. All the data in the figure was derived in the authors lab and we are not sure where the reviewer gets the notion that the data is from another paper. There is nothing in the figure legend to suggest this. The purpose of the figure is to show that there are changes in keratin expression consistent with migration irrespective of treatment. We can show day 7 data, but it does not add any additional information that day 5 does not provide with regards to expression of K17, the purpose of the figure.

4). Figure 7 author claims that they have calculated the thickness of epithelium and percentage of re-epithelization but no data is provided. However figure 7 histological images showed in improvement in treated group. 

The data is provided, it is shown in figure 5. We have changed the figure legend in figure 7 where some aspects were left in as an oversight.

5). Figure 8 why data for day is given and why not day 5. There is no quantification and images show no improvement in healing. 

We were asked by the editor to perform this staining. Although not quantified, the images clearly show in a qualitative manner that the presence of galectin-3, particularly in an in endogenous form, increases the collagen content in areas where cells are migrating in from under the intact skin. Proper quantification would be performed using tensile testing of the skin, but we do not have the infrastructure in our laboratory to do this.

6). Figure 9B show positive cells per cell cm-2, what this means. How authors analyzed microscopic images in cm. 

We apologise for this error. It is not cm2. Cells were quantified in a region of interest size of 200 x 200 µm (40,000 µm2). This has been made clear in the manuscript

7). Figure 11, 12, and 13 are repeated from figure 7, 8, and 9. No new information are provided. 

With all due respect to the reviewer, this is not correct assessment at all. The data in figures 11, 12 and 13 are a separate cohort of animals and there is no overlap with the data shown in figures 7, 8, and 9. Indeed, the tested conditions in the mice are different.

8). There is no novelty in this work, it is repetition of already established fact. 

We strongly disagree with the reviewer in this regard. The reviewer cites no references to support their claim and as such we cannot address this any further than we did after review 1. We addressed this comment in revision 1 to which the reviewer did not respond in revision 2. We respectfully feel that because something is not significant does not make it irrelevant in science, especially in the case of galectin-3. The fundamental goal of our study was to assess whether exogenous delivery of galectin-3 can modulate the cellular processes described for the lectin in vitro. Our results described in this study and that of others suggests strongly that it does not necessarily function in vivo as it does in vitro, and this is an important finding to disseminate to the scientific community. While this study may not have significant data in every figure, the findings are of significance because it shows that how a protein is delivered is an important determinant of its in vivo effects. Moreover, it shows that exogenous delivery of the lectin is not sufficient in vivo to influence healing, and it is likely the expression of galectin-3 by resident cells and its intracellular localization that drives it roles in immunity and re-epithelialization.

            Many studies have been published on galectin-3 which demonstrate effects of the lectin on cell function in vitro, but as studies are moved in vivo, many of these effects are not observed. For example, galectin-3 has been implicated in initiating macrophage polarization into the wound healing related M2 phenotype in vitro (MacKinnon et al. 2008). However, the key aspect in these studies is that up-regulation of galectin-3 expression is a feature of the alternative macrophage phenotype itself and it is the macrophages that release galectin-3. Galectin-3 is highly expressed in M2 macrophages (Novak et al. 2012) and has been shown to play a major role in phagocytosis of opsonized cells (Sano et al. 2003). However, in the study by Sano and colleagues, they demonstrated that genetic deletion of galectin-3 cannot be recovered by addition of exogenous galectin-3. Specifically, they found that "additional experiments indicate that extracellular galectin-3 does not contribute appreciably to the phagocytosis-promoting function of this protein." Karlsson and colleagues further investigated this in vitro using neutrophils, where addition of exogenous galectin-3 in vitro enhanced clearance of neutrophils by monocyte-derived macrophages (Karlsson et al. 2009). In contrast, our study published in the Journal of Investigative Dermatology on excisional wound healing in the galectin-3 knockout mouse demonstrated that deletion of galectin-3 had no effect on macrophage polarization or phenotype (Walker et al, J Invest Dermatol. 2016 May;136(5):1042-1050). Specifically, we showed that genetic deletion of galectin-3 does not affect the immune cell infiltrate at 3 days post wounding or the overall composition of the wounds by 7 days post wounding. Our data in figure 12 in this current manuscript again demonstrates that the addition of exogenous galectin-3 does not significantly influence macrophage polarization, consistent with our analysis of the knockout mouse. Our data in figure 9 shows that at lower concentrations, exogenous galectin-3 does increase M2 macrophages suggesting that it is highly concentration dependent. We conclude from this work, that galectin-3 can alter immune cell function when cells are manipulated in vitro, but when the same processes are assessed in vivo such as in excisional skin healing model systems, the effects of galectin-3 are not as easily observed. Our data presented in the manuscript is in agreement with other data published by other groups.

            Several studies have also implicated galectin-3 in fibrosis, including hepatic (Henderson et al. 2006; Jiang et al. 2012), renal (Henderson et al. 2008) and lung (MacKinnon et al. 2012) fibrosis, suggesting that it may play a role in the fibrotic proliferative phase of wound healing as well. With respect to the lung, pulmonary fibroblasts isolated from galectin-3 knockout animals displayed an impaired myofibroblast differentiation and reduced type I collagen synthesis in vitro (MacKinnon et al. 2012). However, this is likely highly tissue specific as we demonstrated that galectin-3 knockout dermal fibroblasts do differentiate into myofibroblasts and show no change in collagen synthesis versus wild-type cells (Walker et al, J Invest Dermatol. 2016). We further show this in vivo where galectin-3 knockout mice and wild-type mice show no difference in the fibrotic/proliferative phase of healing. Moreover, in figure 11B, our data demonstrates that exogenous galectin-3 can increase epithelial tongue length, which is consistent with our study in knockout mice (Walker et al, J Invest Dermatol. 2016 May;136(5):1042-1050) and that of Liu et al (Liu et al, J Invest Dermatol, 132 (2012), pp. 2828-2837).

            With respect to whether galectin-3 can modulate skin healing in an exogenous form, there are two previous studies. The effect of galectin-3 has been performed in rats in vivo, albeit with much larger wounds (1 cm diameter) than we used in our study (DvoÅ™ánková et al, Cells Tissues Organs. 2011;194(6):469-80). In this study, no effect of galectin-3 was observed. A second study observed a possible role for galectin-3 in excisional repair in rats, specifically that addition of galectin-3 increased collagen expression and the tensile strength of the tissue (Gal et al, Mol Med Rep. 2021 Feb; 23(2): 99). Our data presented in this current manuscript found no effect of galectin-3 on ECM production by dermal fibroblasts in a scaffold form (Figure 4), specifically fibronectin. It also had no effect on fibroblast proliferation, which agrees with our in vivo findings in the knockout mouse (Walker et al, J Invest Dermatol. 2016). We have re-evaluated collagen production in the wounds as recommended by the editor which is presented in figure 8, showing that there is a qualitative increase in collagen deposition in the presence of the galectin-3 scaffolds, which was then confirmed in figure 12 in completely different mice with increased concentrations of exogenous galectin-3. We are now going to re-evaluate collagen production in the galectin-3 knockout mice using more sensitive techniques than Masson’s trichrome and hydroxyproline assays.

The data is statistically meaningful, has appropriate power, and was analyzed using appropriate methods. Where the data is not significant versus controls is likely because galectin-3 delivered in an exogenous form or in a scaffold does not influence the processes in vivo that have been suggested by in vitro studies. Our work is consistent with other data published in the literature with respect to galectin-3 in a variety of different tissue systems.

Round 2

Reviewer 2 Report (Previous Reviewer 2)

Comments and Suggestions for Authors

Accepted in current form. Revision is satisfactory 

Author Response

We thank the reviewer for their support and insightful comments in editing the manuscript. We think the manuscript to be significantly improved based on their helpful comments.

This manuscript is a resubmission of an earlier submission. The following is a list of the peer review reports and author responses from that submission.

Round 1

Reviewer 1 Report

Comments and Suggestions for Authors

The authors have presented their work in a systematic way. However, the following queries should be addressed.

1. In figure 5, why day 2 data is negative. It is essential to repeat the experiment.

2. units representation across the manuscript should be homogeneous

3. The discussion part should be more stronger by comparing the reported literature. The electrospun material should be compared with other reported materials such as membranes or hydrogels. The following references may be useful a) https://doi.org/10.1016/j.jcis.2023.12.156  b. https://doi.org/10.3390/membranes13010102

4. The limitations and future studies should be indicated on the conclusion part

Comments on the Quality of English Language

Spell check should be done

Author Response

Reviewer 1

  1. In figure 5, why day 2 data is negative. It is essential to repeat the experiment.

In certain instances, the initial wound size created can get larger at earlier timepoints due to the loose nature of the skin. However, the actual wound sizes are not significantly different. This has been previously reported in planar measurements of incisional wounds in mice, with an increase in area over the first 2 days of healing "indicating variable expansion of the wound margins" (Ansell et al, Wound Repair Regen. 2014 Mar-Apr;22(2):281-7). Furthermore, planimetry of excisional wounds closure is not as accurate as histological assessment. Planimetry measures only 2D of the surface of the wound, while histology gives assessment of both width and depth, which is more accurate, and histology of the wounds is our primary method we focused on in this manuscript. The closure kinetics data is a measure that is primarily driven by contraction of the wounds and significance is associated more with proteins that directly reduce contraction rather than re-epithelialization. This is evident in our analysis of the influence of periostin deletion on excisional wound healing (Elliott et al, Journal of Cell Science 2012 Jan 1;125(Pt 1):121-32).

  1. units representation across the manuscript should be homogeneous

We apologise for this oversight and all units and measurements have been standardized across the manuscript.

  1. The discussion part should be more stronger by comparing the reported literature. The electrospun material should be compared with other reported materials such as membranes or hydrogels. The following references may be useful a) https://doi.org/10.1016/j.jcis.2023.12.156  b. https://doi.org/10.3390/membranes13010102

We have specifically attempted to focus our discussion with relation to galectin-3 as a potential adjunct therapy to alter inflammation and re-epithelialization. Respectfully, we cannot review all the literature related to skin healing and biomaterials development. Instead we have attempted to focus on the biological activities of galectin-3, potential delivery mechanisms and how our study aligns with the previous research on galectin-3 and skin repair.

  1. The limitations and future studies should be indicated on the conclusion part

We have added a section on study limitations and future studies that should be performed.

Reviewer 2 Report

Comments and Suggestions for Authors

 1.      The study contains detailed analysis associated with gelatin/galectin-3 scaffold effect on wound healing process using in vitro cell based assays, mass spectrometry, tissue histology etc. However, the discussion is restricted to few references mainly focusing on corneal wound healing or knockout galectin study. It does contain other referral studies describing the reasons like why galectin-3 did not show any promoting effect on wound closure even though it has been reported to promote re-epithelization and M1 to M2 transition of macrophage.

2.      Mainly discussion part has some lacking, use more references to justify your findings with different aspects associated with galectin-3 at molecular and cellular level.

3.      In introduction where the role of galectin associated with matricellualr proteins is mentioned, do name those particular MPs with reference studies. Also break the sentence into two as it is long enough. The sentence is as follows:

 “During the skin wound healing process, MPs act spatially and temporally to control specific cell behaviours and of the defined MPs, galectins have received focus due to their influence of inflammatory processes.” (line#52-55) 

4.      What does fiber size means? As fiber diameter is mentioned as fiber character then it is better to use the term “fiber length” instead of size to define the character of fiber length.

5.      In result section as it is stated, “no observable differences in the immunoreactivity for fibronectin was evident between the scaffolds at both of the time points examined.”, do mention that what would be the differences in observations that would confirm the immunoreactivity or is any conformational analysis performed to detect immunoreactivity? (line # 389-391)

6.      If it is claimed that “Proliferation on scaffolds that is consistent with culture on tissue culture plastic is an important finding, as scaffolds made from other materials, including chitosan and polycaprolactone, have demonstrated reduced rates of proliferation.” then give more than one reference demonstrating reduced rates of proliferation in other scaffolds. (line # 619-621)

7.      In sentence, “Based on the evidence suggesting that the scaffolds were biocompatible in vitro, the influence of the scaffolds on wound closure kinetics in a C57BL/6 mouse model of acute excisional wound healing were investigated.”, do mention the type of scaffold? As not all scaffolds are biocompatible. (line # 627-629)

8.      Do write down full forms of abbreviations like CCN2, EGFR, ALIX.

9.      Authors need to explain N=3, n=3 in figure legend.

10.  Figure 4B results are not significant, Figure 5, 6, 7, 10, 11D-E, and 12 results are not significant. It’s difficult to draw a conclusion from this study.

Few grammatical mistakes that I found are mentioned with sentences in the following points.

11.  In a sentence, “Electrospun gelatin/galectin-3 scaffolds developed had an average fiber diameter of 200 nm, with 83% scaffold porosity approximately and pore diameter of approximately 1.15 μm”, it is suggested to write “Electrospun gelatin/galectin-3 scaffolds had an average fiber diameter of 200 nm” (line # 21-23).

12.  The words in vivo and in vitro should be italic throughout the article.

13.  In sentence, “Interestingly, in macrophages derived from galectin-3 deficient mice, IL-4/IL-13-induced M2 macrophage polarization was inhibited, suggesting that galectin-3 is also modulates phenotypes associated with regenerative macrophage activation”, write “suggesting that galectin-3 also modulates phenotypes” instead of “galectin-3 is also modulates phenotypes.” (line # 71-73).

14.  In sentence, “In addition to regulating inflammatory processes, studies using galectin-3 knockout mice show that keratinocytes exhibit a migratory defect that manifests in the healing of excisional skin wounds, with time to complete re-epithelialization delayed compared to wild-type mice”, it is better to split it into two or rephrase it. (line # 74-77)

15.  Rephrase the sentence, i.e., “This defect related is to deficient epidermal growth factor receptor (EGFR) endocytosis and recycling, which is controlled by cytosolic galectin-3 binding to Alpha-1,3/1,6-mannosyltransferase interacting protein X (ALIX)”, for better understanding. (line # 77-79)

Comments on the Quality of English Language

See comments 

Author Response

Reviewer 2

  1. The study contains detailed analysis associated with gelatin/galectin-3 scaffold effect on wound healing process using in vitro cell based assays, mass spectrometry, tissue histology etc. However, the discussion is restricted to few references mainly focusing on corneal wound healing or knockout galectin study. It does contain other referral studies describing the reasons like why galectin-3 did not show any promoting effect on wound closure even though it has been reported to promote re-epithelization and M1 to M2 transition of macrophage.

As discussed above, we specifically attempted to focus our discussion with relation to galectin-3 as a potential adjunct therapy to alter inflammation and re-epithelialization based on its defined roles from previous studies. We have added a small section on potential reasons why galectin-3 did not function as expected to the discussion.

  1. Mainly discussion part has some lacking, use more references to justify your findings with different aspects associated with galectin-3 at molecular and cellular level.

      We have restructured the discussion and added appropriate references as requested by the reviewer.

  1. In introduction where the role of galectin associated with matricellular proteins is mentioned, do name those particular MPs with reference studies. Also break the sentence into two as it is long enough. The sentence is as follows:

 “During the skin wound healing process, MPs act spatially and temporally to control specific cell behaviours and of the defined MPs, galectins have received focus due to their influence of inflammatory processes.” (line#52-55).

We have added information on other matricellular proteins and reworded the sentence as requested.   

  1. What does fiber size means? As fiber diameter is mentioned as fiber character then it is better to use the term “fiber length” instead of size to define the character of fiber length.

      Fiber size has been clarified. It refers to the diameter of the fiber and we have made this clear in the revised manuscript.

  1. In result section as it is stated, “no observable differences in the immunoreactivity for fibronectin was evident between the scaffolds at both of the time points examined.”, do mention that what would be the differences in observations that would confirm the immunoreactivity or is any conformational analysis performed to detect immunoreactivity? (line # 389-391)

      We did not attempt to quantify the fibronectin immunoreactivity; we used this as a measure to demonstrate that the scaffolds supported fibronectin deposition. In summation, with increasing time, there is an increase in fibronectin deposition by cell on all tested scaffolds.

  1. If it is claimed that “Proliferation on scaffolds that is consistent with culture on tissue culture plastic is an important finding, as scaffolds made from other materials, including chitosan and polycaprolactone, have demonstrated reduced rates of proliferation.” then give more than one reference demonstrating reduced rates of proliferation in other scaffolds. (line # 619-621)

      Additional references have been added as requested by the reviewer.

  1. In sentence, “Based on the evidence suggesting that the scaffolds were biocompatible in vitro, the influence of the scaffolds on wound closure kinetics in a C57BL/6 mouse model of acute excisional wound healing were investigated.”, do mention the type of scaffold? As not all scaffolds are biocompatible. (line # 627-629)

      We apologise for this oversight. The scaffold types have been inserted in the appropriate place in the manuscript.

  1. Do write down full forms of abbreviations like CCN2, EGFR, ALIX.

      We have added the abbreviations as requested. We apologise for this oversight.

  1. Authors need to explain N=3, n=3 in figure legend.

      This information has been added.

  1. Figure 4B results are not significant, Figure 5, 6, 7, 10, 11D-E, and 12 results are not significant. It’s difficult to draw a conclusion from this study.

We agree with the reviewer that the results of the study are not significant, which we theorize is itself very significant for the use of local delivery of galectin-3 to modulate re-epithelialization or macrophage phenotype. Our prior analysis of excisional wound healing in the galectin-3 knockout mouse demonstrated that deletion of galectin-3 only influenced the rate of re-epithelialization, and had no effect on macrophage polarization or phenotype (Walker et al, J Invest Dermatol. 2016 May;136(5):1042-1050). With respect to re-epithelialization, our study agreed with the findings of the study by Liu et al (J Invest Dermatol, 132 (2012), pp. 2828-2837), showing that Galectin-3 is only required for re-epithelialization. Similar studies on the effect of galectin-3 in vivo have been performed in rats, albeit with much larger wounds (1 cm diameter) than we used in our study (DvoÅ™ánková et al, Cells Tissues Organs. 2011;194(6):469-80). In this study, no effect of galectin-3 was observed, although a second study highlighted a potential role for galectin-3 in excisional repair, specifically increasing collagen expression and the tensile strength of the tissue (Gal et al, Mol Med Rep. 2021 Feb; 23(2): 99).

Few grammatical mistakes that I found are mentioned with sentences in the following points.

The following grammatical errors have all been changed. We thank the reviewer for their considerable time to read the manuscript in depth.

  1. In a sentence, “Electrospun gelatin/galectin-3 scaffolds developed had an average fiber diameter of 200 nm, with 83% scaffold porosity approximately and pore diameter of approximately 1.15 μm”, it is suggested to write “Electrospun gelatin/galectin-3 scaffolds had an average fiber diameter of 200 nm” (line # 21-23).

  1. The words in vivo and in vitro should be italic throughout the article.

  1. In sentence, “Interestingly, in macrophages derived from galectin-3 deficient mice, IL-4/IL-13-induced M2 macrophage polarization was inhibited, suggesting that galectin-3 is also modulates phenotypes associated with regenerative macrophage activation”, write “suggesting that galectin-3 also modulates phenotypes” instead of “galectin-3 is also modulates phenotypes.” (line # 71-73).

  1. In sentence, “In addition to regulating inflammatory processes, studies using galectin-3 knockout mice show that keratinocytes exhibit a migratory defect that manifests in the healing of excisional skin wounds, with time to complete re-epithelialization delayed compared to wild-type mice”, it is better to split it into two or rephrase it. (line # 74-77)

  1. Rephrase the sentence, i.e., “This defect related is to deficient epidermal growth factor receptor (EGFR) endocytosis and recycling, which is controlled by cytosolic galectin-3 binding to Alpha-1,3/1,6-mannosyltransferase interacting protein X (ALIX)”, for better understanding. (line # 77-79)

Reviewer 3 Report

Comments and Suggestions for Authors

The manuscript deals with Exogenous Galectin-3 Delivery for Modulating 2 Macrophage
Polarization. This manuscript is poorly organized, and many significant issues must be solved. My significant comments are as follows:

The mechanism is missing. No adequate studies have been done for this research work.

Similar study was done before. What was the novelty of the study?

Authors have prepared the materials via electrospinning how ever chemical structure of the material was unknown.

Reduce the repeating words.

Check the second affiliation, country and other information is missing.

Introduction have too much information and lengthy. Please shorten it as per novelty of this study. Delete the too general information.

Instrument model name and other information is missing for ex: line 126.

Line 184, make a space between number and oC. Check other places in the manuscript.

In vitro and in vivo should be italics in the manuscript.

Tables format are wrong as per the journal guidelines.

Fig. 10 a scale bar is missing.

Comments on the Quality of English Language

Extensive editing of English language required

Author Response

Reviewer 3

  1. The mechanism is missing. No adequate studies have been done for this research work.

Similar study was done before. What was the novelty of the study?

Authors have prepared the materials via electrospinning however chemical structure of the material was unknown.

The focus of the study was to assess two different mechanisms of delivery of galectin-3 in wound healing. We selected two readouts that have been described as modulated by galectin-3 in vitro. We ourselves had previously published an assessment of wound healing in the galectin-3 knockout mouse and established that deletion of galectin-3 only effected the rate of re-epithelialization, but not macrophage polarization (Walker et al, J Invest Dermatol. 2016 May;136(5):1042-1050). With respect to re-epithelialization, our study agreed with the findings of the study by Liu et al (J Invest Dermatol, 132 (2012), pp. 2828-2837). As highlighted by the reviewer, similar work has been performed in rats, albeit with much larger wounds (1 cm diameter) than we used in our study (DvoÅ™ánková et al, Cells Tissues Organs. 2011;194(6):469-80). In this study, not effect of galectin-3 was observed, although a second study highlighted a potential role for galectin-3 in excisional repair, specifically increasing collagen expression and the tensile strength of the tissue (Gal et al, Mol Med Rep. 2021 Feb; 23(2): 99). At the time the Gal et al paper was published, we had already been investigating galectin-3 as a potential modulator of inflammation and re-epithelialization. Respectfully, we feel our paper contributes significantly to the literature, confirming that the role of galectin-3 in skin repair is possibly context dependent.

  1. Reduce the repeating words.

Check the second affiliation, country and other information is missing.

Introduction have too much information and lengthy. Please shorten it as per novelty of this study. Delete the too general information.

All affiliations have been updated and the manuscript has been carefully proofread/ The introduction has been shortened as requested with superfluous information removed.  

  1. Instrument model name and other information is missing for ex: line 126.

Line 184, make a space between number and oC. Check other places in the manuscript.

In vitro and in vivo should be italics in the manuscript.

Tables format are wrong as per the journal guidelines.

The instrument model number has been added and italics used as requested. As far as we can read, the table format meet journal requirements.

  1. Fig. 10 a scale bar is missing.

The scale bar has been added to the figure. We apologize for this oversight.

Round 2

Reviewer 2 Report

Comments and Suggestions for Authors

The author addressed the minor issues in the revised version. However, the major concern "Figure 4B results are not significant, Figures 5, 6, 7, 10, 11D-E, and 12 results are not significant. It’s difficult to draw a conclusion from this study." is not addressed. 

Overall the manuscript's 6-figures results are not significant. Therefore, in my opinion, this manuscript couldn't be published. 

Comments on the Quality of English Language

Fine 

Reviewer 3 Report

Comments and Suggestions for Authors

Authors improved the manuscript.

Comments on the Quality of English Language

Minor checking needed.